# T-norm Selection for Object Detection in Autonomous Driving with Logical Constraints

**Thomas Eiter**[1]    **Nelson Higuera Ruiz**[1]    **Katsumi Inoue**[2]    **Sota Moriyama**[2,3]

[1]Vienna University of Technology, Austria
[2]National Institute of Informatics, Japan
[3]The Graduate University for Advanced Studies, SOKENDAI, Japan
{thomas.eiter,nelson.ruiz}@tuwien.ac.at,  {inoue,sotam}@nii.ac.jp

## Abstract

Integrating logical constraints into object detection models for autonomous driving (AD) is a promising way to enhance their compliance to rules and thus increase the safety of the system. In this, t-norms have been utilized to calculate the constrained loss, i.e., the violations of logical constraints as losses. While prior works have statically selected few t-norms, we conduct an extensive experimental study to identify the most effective choices, as suboptimal t-norms can lead to undesired model behavior. For this, we present MOD-ECL, a neurosymbolic framework that implements a wide range of t-norms and can use them in an adaptive manner, with an algorithm that selects well-performing t-norms during training and a scheduler that regulates the impact of the constrained loss. We evaluate its effectiveness on the ROAD-R and ROAD-Waymo-R datasets for object detection in AD with attached common-sense constraints. Our results show that careful selection of parameters is crucial for good behavior of the constrained loss and that our framework allows us to obtain not only lower constraint violation but in some cases also an increase in detection performance. Furthermore, our methods allow fine control over the tradeoff between accuracy and violation.[1]

## 1 Introduction

Object detection models [33] have become essential for autonomous driving (AD) applications [17] due to their efficiency and accuracy [34], and a variety of datasets have been created to evaluate these models [2, 18, 27, 9]. However, standard data-driven models do not consider formal symbolic knowledge, and they often fail to satisfy logical constraints that reflect real-world rules and relationships between detected objects; for example, in an AD scenario pedestrians usually should not be detected on top of a moving vehicle. The consistent satisfaction of this and similar kinds of constraints is crucial for creating safe and reliable autonomous driving systems.

Neurosymbolic methods [5] have emerged as a promising solution to this problem, supporting different ways to integrate knowledge into deep learning models, cf. [4, 8]. An effective approach for autonomous driving is to embed logical constraints into loss functions using t-norms [1], with the constrained loss [7] as a representative. The latter is aggregated with a regularization term $\lambda$ to the loss functions of a neural network, driving the model to simultaneously learn satisfying the logical constraints and solving the detection task.

Recent research has considered the inclusion of *requirements* for AD scenarios expressed as logical constraints in datasets [12, 15]. It appeared that using constrained loss can increase constraint satisfaction as wells as model performance of the overall AD system. On the other hand, the results

---

[1]All our code is contained in an online repository: `https://github.com/pudumagico/MOD-ECL/`

39th Conference on Neural Information Processing Systems (NeurIPS 2025).

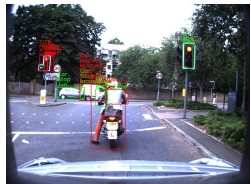
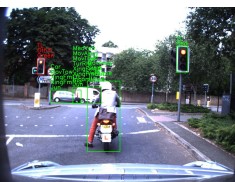
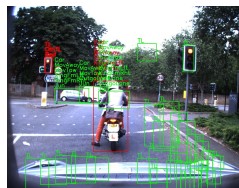
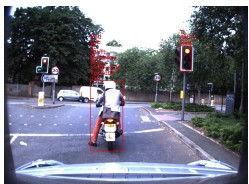

| (a) w/o Constrained Loss, i.e., $\lambda = 0$. | (b) w/ Constrained Loss, using Product t-norm, $\lambda = 50$. | (c) w/ Constrained Loss, using Product t-norm, $\lambda = 1000$. | (d) w/ Constrained Loss, using Łukasiewicz t-norm, $\lambda = 50$. |
|---|---|---|---|

Figure 1: Four images showing the inference results after 10 epochs of training the model *MOD-ECL* (see Section 4), with labels associated to bounding boxes above the top-left corner. Bounding boxes in **green** indicate satisfaction of logical requirements and in **red** violations. Compared to the baseline in Fig. 1a (regularization term $\lambda = 0$), we highlight that using constrained loss with product t-norm can yield higher detection performance and lower requirement violations, as shown in Fig. 1b. However, improper $\lambda$ respectively t-norm selection can severely hinder the learning and constraint satisfaction, as shown in Figs. 1c and 1d.

showed that the choice of a t-norm and regularization term $\lambda$ may significantly affect constraint satisfaction as well as detection performance, leading to undesired results as in Figure 1. Furthermore, there is a plethora of t-norms [28], but only few of them have been considered so far. This poses the challenge of finding a best t-norm efficiently, which should achieve high constraint satisfaction while not degrading detection performance.

To overcome these issues, we make the following contributions:

- First, to explore a possibly large number of t-norms, we introduce an adaptive algorithm to dynamically explore and select the one that reduces some score by the most during training. This adaptive approach also enables the user to measure which t-norms are used the most, and thus to focus on them for their particular application, with asserted theoretical properties.

- Secondly, to control the effect of the constrained loss, we introduce a *scheduler* that dynamically adjusts $\lambda$ during training, attempting to separate stages where the model is focused on learning object detection or on constraint satisfaction, where intuitively, the former should happen in early stages.

- Thirdly, we present MOD-ECL, a novel neurosymbolic framework that handles a wide range of t-norms, with functions that automatically search t-norms and allow for parameter tuning in order to optimize model performance, based on adaptive t-norm selection and $\lambda$ scheduling.

- Finally, we conduct an extensive experimental evaluation of the approach on two backbone object detection models deployed to two AD datasets augmented with requirements as logical constraints, viz. ROAD-R[2], an extension of ROAD from the ROAD Challenge[3] with 243 requirements, and ROAD-Waymo-R, a subset of ROAD-Waymo from the ROAD++ Challenge[4] with 251 requirements. In the evaluation, we study the choice of t-norms as well as variation of the regularization $\lambda$. The results reveal that the choice and variation can considerably affect the outcome. Some t-norms yield higher detection performance while failing to simultaneously increase constraint satisfaction, and vice versa. However, we found that there are certain scenarios where both metrics improve drastically or offer a reasonable tradeoff (see Section 5.4 for detailed discussion).

Overall, our results advance the integration of logical constraints in practical AD scenarios, providing a framework and tools to assist exploration of different options in a controlled manner. Furthermore, they provide insight into the behavior of a range of t-norms for this task.

## 2 Related Work

Neurosymbolic systems consist of neural and logical modules that interact with each other to solve some task [20]. An interesting question is how to use the logical part to enhance the learning

---

[2]https://sites.google.com/view/road-r
[3]https://sites.google.com/view/roadchallangeiccv2021
[4]https://sites.google.com/view/road-plus-plus

capabilities of the neural one by modifying the loss function. Many systems [32, 19, 30] rely on model counting techniques for calculating the *semantic loss* [31], whose use proved to be successful in reducing the amount of data needed to reach a level of accuracy and making neural networks comply with a set of constraints. Unfortunately, due to the high complexity of model counting [13], this method has only been fruitfully applied to small scale problems such as arithmetic operations over MNIST [6] or sudoku.

A different approach called *Semantic Based Regularization* [7] (essentially the constrained loss) bypasses this issue by considering the single world with the highest predictions from the neural network and the use of t-norms to calculate the violations. This allows one to handle the scale of real world problems such as AD where integrating logical constraints is a promising way to improve compliance with formal background knowledge. Diligenti et al. considered the Product, Gödel and Łukasiewicz t-norms in their first approach and experimented on an object detection task [29], where they showed that adding rules can lead to gains in accuracy and F1 score. Their research was continued in [10], where they expanded the theory of t-norms to include generators and considered also parameterized t-norms such as the Schweizer–Sklar and Frank t-norms. They showed that by solely using t-norm based learning, their model was able to learn the MNIST dataset with high accuracy. In a further paper [11], the authors proposed *Deep Fuzzy Logic (DFL)*, a theoretical framework for constrained learning maintaining the aforementioned t-norms. They compare with various baselines and show that using DFL improves $\approx 5\%$ over a neural network over the CiteSeer [25] dataset. While these works developed the idea of integrating t-norms, they fall short of addressing a practical real-world use case such as autonomous driving.

In the AD domain, the recent ROAD-R dataset [12] provides a structured framework to evaluate a model's ability to comply with logical constraints. The authors showed that existing state-of-the-art models frequently violate these constraints, with more than 90% of predictions being non-admissible, i.e., they violate some constraint. To address this, they proposed incorporating constrained loss, constrained output (filtering compliant predictions only), or both. A follow-up paper [26] addressed the computational challenges of integrating logical constraints via t-norms into deep learning models. The authors proposed a memory-efficient implementation of t-norm-based loss functions that reduces the usually prohibitive memory requirements of these methods. They conducted experiments with the ROAD-R dataset to demonstrate that incorporating standard t-norms improves object detection performance, particularly in scenarios with limited labeled data, achieving up to a 3.95% improvement when only 20% of labeled data is available. Furthermore, they showed that applying t-norm-based constraints to both labeled and unlabeled data yields further performance gains. The work is continued in ROAD-Waymo-R [15], where a follow-up AD dataset with requirements, the experiments showed that using standard t-norms in the constrained loss improves performance over the baseline I3D model [3]. The Gödel t-norm offered a performance boost highlighting the potential of domain knowledge integration during training to enhance complex perception tasks.

## 3 Background

### 3.1 ROAD-R and ROAD-Waymo-R Datasets

The ROAD-R dataset extends the ROAD dataset by incorporating common sense logical constraints in the AD context as requirements. The original ROAD dataset, built on top of the Oxford RobotCar Dataset [18], contains a total of 41 classes and 122K frames from 22 annotated videos, where each of them is about 8 minutes long. The labels are in the format of tuples $(o, l, a)$ where $o$ is an object class, $l$ is a location class, and $a$ is an action class. For example, the detection (Pedestrian, Crosswalk, Walking, PushObj) represents a pedestrian located on a crosswalk, performing the activities of walking and pushing an object. Note that each component can include multiple labels of the same type, making multi-label classification necessary.

ROAD-R introduces a set of 243 logical requirements expressed in propositional logic. These constraints cover relationships between objects, locations and actions; for instance, the statement "vehicle lane cannot be a parking lot" says that two locations are different. The logical constraints are disjunctions of literals which we represent as sets of literals, e.g., $\neg$VehicleLane $\vee$ $\neg$ParkingLot as $\{\neg$VehicleLane$, \neg$ParkingLot$\}$.

The ROAD-Waymo [15] dataset, built on top of the Waymo Open Dataset [21], is considerably more comprehensive and challenging than any existing AD benchmark, including ROAD. It contains 198k

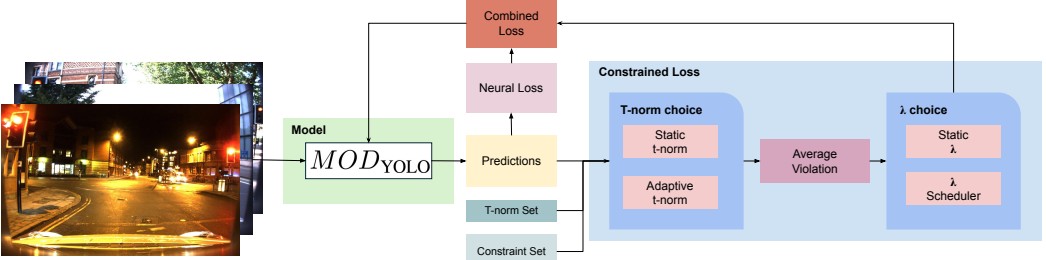

Figure 2: *MOD-ECL* framework instantiated for AD with MOD$_{\text{YOLO}}$. The model generates predictions, which are evaluated against a constraint set using t-norms. A choice between static and adaptive t-norms can be made, with the latter dynamically adjusting during training. The regularization term ($\lambda$) can be either static or adjusted using a scheduler. The final combined loss incorporates both the neural loss and the constrained loss, balancing predictive accuracy with requirement consistency.

frames from 5 hours of footage and labels are in the same tuple format $(o, l, a)$ as in ROAD. There are 58 classes, and some of them differ from the ones in ROAD. It includes a set of 251 constraints, modeled keeping the central common parts with the ROAD-R constraints, removing constraints that do not apply, and adding new ones that apply.

## 3.2 T-norms

T-norms (triangular norms) [16] are mathematical functions used in fuzzy logic to generalize the concept of logical conjunction (AND). Given two values $a, b \in [0, 1]$, a t-norm $T$ combines them to produce a value $T(a, b)$ that represents their joint degree of truth, obeying monotonicity, i.e., $T(a, b) \leq T(a', b')$ if $a \leq a'$ and $b \leq b'$, and is viewed as a binary operator commutative, associative, and has 1 as neutral element. T-norms enable the integration of logical constraints into neural networks, as they allow for the relaxation of hard constraints into differentiable forms that can be optimized during training.

While t-norms are defined as binary functions, we sometimes want to apply them to constraints with more than two variables. The presented t-norms can be naturally extended to multiple variables, yet one should be careful when using more complex expressions. We apply a t-norm $T$ to a set $fv = \{a_1, \ldots, a_n\}$, $n \geq 2$, of fuzzy values iteratively, i.e., we define $T(fv)$ by $T(fv) = T(a_1, a_2)$ if $n = 2$ and $T(fv) = T(T(\{a_1, \ldots, a_{n-1}\}, a_n)$ otherwise; by commutativity and associativity of $T$, the enumeration of $fv$ is immaterial. An assignment of values to a constraint $c$ can be defined as $c[v] = c \mid_{x_i \mapsto v_i}$, where $x = x_1, \ldots, x_n$ are the variables in $c$, $v = v_1, \ldots, v_n$ is a list of values, and $x_i \mapsto v_i$ denotes the substitution of each variable $x_i$ by $v_i$, resulting in the modified formula $c[v]$.

## 3.3 Constrained Loss

Constrained loss [7, 12] is a technique to enforce logical constraints during the training of neural networks. It modifies the traditional loss function by adding a penalty term that measures the degree to which the model's predictions violate predefined constraints. This penalty can be expressed as

$$\mathcal{L}_{\text{comb}} = \mathcal{L}_{\text{nn}} + \lambda \mathcal{L}_{\text{cl}}, \tag{1}$$

where $\mathcal{L}_{\text{nn}}$ is the standard loss (e.g., cross-entropy), $\mathcal{L}_{\text{cl}}$ is the constraint violation loss [12], and $\lambda$ balances task performance and constraint satisfaction; $\mathcal{L}_{\text{comb}}$ is their regularized combination. Considering that negations are computed using *strong negation*, which is formally defined as $1 - a$ given $a \in [0, 1]$, we define the constrained loss term inspired by [26] as

$$\mathcal{L}_{\text{cl}} = \frac{1}{n} \frac{1}{|\mathcal{B}|} \sum_{j=1}^{m} \sum_{i=1}^{n} T\left(c_i[1 - b_j]\right), \tag{2}$$

where $T$ is the t-norm for evaluating constraint satisfaction, $C = \{c_1, \ldots, c_n\}$ is the set of constraints, and $\mathcal{B} = \{b_1, \ldots, b_m\}$ is the set of detections, where each $b_j$ contains predicted confidence scores per label and $1 - b_j$ denotes their strong negation.

**Example** (Constrained Loss Computation). *Consider the rule: "Only pedestrians or cyclists can wait to cross" This can be written in propositional logic as:*

$$Pedestrian \lor Cyclist \lor \neg Wait2X.$$

| Framework | Datasets | Models | T-norm library | Adaptive T-norm selection | $\lambda$-scheduler |
|---|---|---|---|---|---|
| MOD-CL | 1 | 1 | 3 t-norms | ✗ | ✗ |
| *MOD-ECL* | 2 | 2 | 12 t-norms | ✓ | ✓ |

Table 1: Key differences between MOD-ECL and the prior MOD-CL framework.

*Assume the detector assigns the following confidence values:*

$$\texttt{Pedestrian} = a = 0.9, \quad \texttt{Cyclist} = b = 0.2, \quad \texttt{Wait2X} = c = 0.7.$$

*Using the Product t-norm, $T_P(x,y) = x \cdot y$, the corresponding constrained loss is:*

$$T_P(a, b, 1-c) = 0.1 \times 0.8 \times 0.7 = 0.056$$

*Hence, the model incurs a loss of $0.056$ for this rule under the current predictions.*

## 4 MOD-ECL

Prior to introducing our novel algorithms, we present in this section our framework *MOD-ECL*, which stands for Multilabel Object Detection with Enhanced Constrained Loss. It is a framework for integrating constraint knowledge into multilabel object detection models through constrained loss, leveraging on MOD-CL [22], a framework that produced the winning model for Task 2 and the 3rd-best model of Task 1 of the NeurIPS 2023 ROAD-R Challenge [5]. *MOD-ECL* comprises a collection *MOD-ECL* $= (\mathcal{D}, \mathcal{M}, C, \mathcal{T})$ of a dataset $\mathcal{D}$, a multilabel object detection model $\mathcal{M}$, a set $C$ of constraints, and finally a set $\mathcal{T}$ of t-norms. It is assumed that $\mathcal{M}$ returns, given an input image, a set $\mathcal{B}$ of bounding boxes as output, where each bounding box $b_j \in \mathcal{B}$ can have multiple labels.

*MOD-ECL* integrates the background knowledge of the constraints $C$ for the dataset $\mathcal{D}$ into the neural network model $\mathcal{M}$ using the loss function shown in Equation (1). Moreover, for the calculation of the constrained loss in Equation (2), it uses the t-norms in the set $\mathcal{T}$. The latter may consist of a single t-norm (*static t-norm*), or of multiple t-norms that will be adaptively selected as considered below (*adaptive t-norm*). The pipeline of *MOD-ECL* for AD is shown in Figure 2.

**Comparison to MOD-CL.** *MOD-ECL* extends the previous MOD-CL framework for constraint-aware object detection through three key innovations, the differences highlighted in Table 1. First, it introduces a significantly broader library of t-norms, expanding from 3 to 12, including both non-parameterised and parameterised families. Second, it proposes an adaptive t-norm selection mechanism, which frames the problem as a multi-armed bandit and dynamically identifies well-performing t-norms during training. Third, a novel $\lambda$-scheduler is introduced to modulate the influence of the constrained loss over time, ensuring that the model initially focuses on detection performance before progressively enforcing constraint satisfaction. These additions result in a robust, modular, and fully automated training pipeline that improves constraint compliance with minimal degradation—and occasional improvements—in detection accuracy.

### 4.1 MOD_YOLO: Multi-label Object Detection with YOLO

In our work, we specifically focus on an instantiation of *MOD-ECL* for the ROAD-R and ROAD-Waymo-R datasets, using the constraints given in prior works. In this section, we particularly focus on the object detection model $\mathcal{M}$ that is used.

Specifically, we use $\mathcal{M} = MOD_{YOLO}$ from the MOD-CL framework for the ROAD-R Challenge dataset. $MOD_{YOLO}$ is a multilabel extension of YOLO models v8n and 11n [14][6]. Both the training and the inference procedure of the YOLOv8 model were modified: In training, n-hot vector representations of all labels were used as ground truth labels to allow for training on multilabel instances. In inference, Non-Maximum Suppression (NMS) [24] is used in a specific manner: object specific labels were used. In the case of ROAD, each object is given only one agent label; we perform NMS based on agent labels.

---

[5] https://github.com/sotam2369/MOD-CL
[6] https://yolov8.com, https://docs.ultralytics.com/models/yolo11

| $T$ | Definition | $T$ | Definition |
|---|---|---|---|
| $T_{\mathrm{P}}$ | $a \cdot b$ | $T_{\mathrm{SS}}$ | $\begin{cases} \min(a,b), & p = 0 \\ \max(0, (a^p + b^p - 1)^{1/p}), & \text{otherwise} \end{cases}$ |
| $T_{\mathrm{G}}$ | $\min(a,b)$ | $T_{\mathrm{H}}$ | $\dfrac{a \cdot b}{p + (1-p) \cdot (a + b - a \cdot b)}$ |
| $T_{\mathrm{L}}$ | $\max(a + b - 1, 0)$ | $T_{\mathrm{F}}$ | $\begin{cases} a + b - 1, & p = 1 \\ \log_p\left(1 + \dfrac{(p^a - 1)(p^b - 1)}{p - 1}\right), & \text{otherwise} \end{cases}$ |
| $T_{\mathrm{D}}$ | $\begin{cases} 1, & \text{if } a = 1 \text{ or } b = 1 \\ 0, & \text{otherwise} \end{cases}$ | $T_{\mathrm{Y}}$ | $\max\left(0, 1 - ((1-a)^p + (1-b)^p)^{1/p}\right)$ |
| $T_{\mathrm{HP}}$ | $\begin{cases} \dfrac{a \cdot b}{a + b - a \cdot b}, & a, b \neq 0 \\ 0, & \text{otherwise} \end{cases}$ | $T_{\mathrm{SW}}$ | $\dfrac{a + b - 1 + p \cdot a \cdot b}{1 + p}$ |
| $T_{\mathrm{NM}}$ | $\begin{cases} \min(a,b), & a + b > 1 \\ 0, & \text{otherwise} \end{cases}$ | $T_{\mathrm{AA}}$ | $\exp\left(-((-\log a)^p + (-\log b)^p)^{1/p}\right)$ |

Table 2: T-norms implemented in MOD-ECL. Symbols correspond to: $T_{\mathrm{P}}$ (Product), $T_{\mathrm{G}}$ (Gödel), $T_{\mathrm{L}}$ (Łukasiewicz), $T_{\mathrm{D}}$ (Drastic), $T_{\mathrm{HP}}$ (Hamacher Product), $T_{\mathrm{NM}}$ (Nilpotent Minimum), $T_{\mathrm{SS}}$ (Schweizer-Sklar), $T_{\mathrm{H}}$ (Hamacher), $T_{\mathrm{F}}$ (Frank), $T_{\mathrm{Y}}$ (Yager), $T_{\mathrm{SW}}$ (Sugeno-Weber), $T_{\mathrm{AA}}$ (Aczél-Alsina).

For t-norm computation, we further prune detections: In YOLOv8, models may output numerous detections, many of which may have a very low confidence score for each possible class. Such detections are considered to be "background" classifications, and we do not want to compute violations from them. Thus, when calculating violations, we look at detections that have at least one label with a confidence score above 0.5, a value that is deemed as leaning towards true in fuzzy logic.

### 4.2 Implemented T-norms

For the instantiation of $\mathcal{T}$, we implement standard t-norms from related literature, namely $T_{\mathrm{P}}$, $T_{\mathrm{G}}$, and $T_{\mathrm{L}}$ alongside numerous new ones, all shown in Table 2 . They are divided into non-parameterized and parameterized t-norms, where the latter offer a degree of flexibility regarding some parameter $p \geq 0$. The semantics of t-norms may vary drastically, which motivates an empirical evaluation on the datasets. In addition to $T_{\mathrm{G}}$, $T_{\mathrm{L}}$, and $T_{\mathrm{P}}$, we implement three more non-parameterized t-norms. $T_{\mathrm{D}}$ is an extreme t-norm that returns 1 if either value is 1 and returns 0 otherwise. $T_{\mathrm{HP}}$ produces a smooth continuous curve, similar to $T_{\mathrm{P}}$. $T_{\mathrm{NM}}$ is a stricter version of $T_{\mathrm{G}}$ that returns 0 whenever the sum of the inputs is at most 1. In addition, parameterized t-norms have a parameter $p \geq 0$ to adjust the behavior, offering some degree of control between strict and relaxed conjunctions. Some of them admit also $p < 0$; we disregard this for simplicity. Depending on $p$, some parameterized t-norms amount to non-parameterized ones; e.g., $T_{\mathrm{H}}$ becomes $T_{\mathrm{HP}}$ whenever $p = 0$ and $T_{\mathrm{SS}}$ coincides with $T_{\mathrm{L}}$ for $p = 1$ and $T_{\mathrm{P}}$ with $p = 1$.

### 4.3 Adaptive Algorithm for T-norm Selection

To select the best t-norm $T^* \in \mathcal{T}$ during training, we use an adaptive algorithm that balances *exploration* (random selection with probability $\beta$) and *exploitation* (choosing the t-norm with the highest score). As shown in Algorithm 1, the score of each t-norm reflects its recent performance in reducing the constrained loss in Equation (2), which measures the degree to which logical constraints are violated. At each iteration, it either explores by randomly selecting a t-norm with probability $\beta$, or exploits by choosing the t-norm with the highest accumulated score (line 5). After selecting a t-norm, the algorithm computes its constrained loss (lines 6). If this is the first time the t-norm is evaluated, the loss is stored without updating the score. Otherwise, the algorithm computes a normalized improvement score (lines 7–11) based on the reduction in constraint loss; this is scaled either by the previous loss or a small constant $\epsilon$ to avoid instability when losses are zero. Finally, the score for the selected t-norm is updated using exponential discounting (lines 12–13), allowing the algorithm to gradually adapt to changing performance trends while preserving historical performance. At the end of the iteration the t-norm with the highest score is returned as the best choice (line 14).

**Algorithm 1** Training with Adaptive T-norm Selection

---

**Require:** Number of batches per epoch $b$, epochs $e$, set of t-norms $\mathcal{T}$, exploration probability $\beta$, discount factor $\delta$, small constant $\epsilon$.
**Ensure:** Trained model `model`
1: `losses, scores` $\leftarrow \{T:0\ \forall T \in \mathcal{T}\}$, `iterations` $\leftarrow b \cdot e$
2: $T^* \leftarrow$ `random.choice`$(\mathcal{T})$                    *// Initialize best t-norm*
3: **for** $i = 1$ **to** `iterations` **do**
4:    `model` $\leftarrow$ `train_model`$(T^*)$              *// Train on current best t-norm*
5:    $T_i \leftarrow \begin{cases} \texttt{random.choice}(\mathcal{T}), & \text{if } \texttt{random}() \leq \beta \\ \arg\max_{T \in \mathcal{T}} \texttt{scores}[T], & \text{otherwise} \end{cases}$    *// Candidate to evaluate*
6:    $cl_c \leftarrow$ `compute_constraint_loss`$(T_i)$
7:    $cl_p \leftarrow$ `losses`$[T_i]$,   `losses`$[T_i] \leftarrow cl_c$
8:    **if** $cl_p = \infty$ **then**
9:       **continue**
10:   **end if**
11:   $n_i \leftarrow (cl_p - cl_c)/\max(\epsilon, cl_p)$            *// Normalized improvement*
12:   $s \leftarrow$ `scores`$[T_i]$
13:   `scores`$[T_i] \leftarrow \begin{cases} n_i, & \text{if } s = 0 \\ \delta\,s + (1-\delta)\,n_i, & \text{otherwise} \end{cases}$
14:   $T^* \leftarrow \arg\max_{T \in \mathcal{T}}$ `scores`$[T]$             *// Update best t-norm*
15: **end for**
16: **return** `model`

---

We can prove for Algorithm 1 that every t-norm is selected infinitely often, ensuring sufficient exploration and furthermore Theorem 1 under common assumptions, such as stationary and bounded scores, guarantees that the adaptive t-norm selection regret, i.e., the number of times a suboptimal t-norm is chosen is bounded. This comes from results in multi armed bandit (MAB) [23] theory.

**Theorem 1** (Regret Bound). *For any suboptimal t-norm $T$ whose expected discounted normalized gain $\mu_T$ is strictly lower than that of the best t-norm $T^*$ (i.e., $\mu_T < \mu_{T^*}$), the expected number of times $T$ is selected by Algorithm 1 satisfies $\mathbb{E}[N_T(n)] \leq \log n/(\beta \cdot \Delta_T^2)$ where $\Delta_T := \mu_{T^*} - \mu_T$ is the expected discounted performance gap.*

In practice, not all assumptions may be fulfilled, but we see in our empirical results that the algorithm stabilizes and chooses one t-norm consistently. Proofs are included in the Appendix.

## 4.4 $\lambda$ Scheduler

After calculating the constrained loss $\mathcal{L}_{\text{cl}}$, we must add it to the neural loss $\mathcal{L}_{\text{nn}}$ and tune the $\lambda$ value (cf. (1)). A similar approach called *warm-up* is presented in [26], but the $\lambda$ remains constant. In contrast, we study the effect of increasing $\lambda$ during training. Intuitively, if $\lambda$ is very high, the constraints will be "hard", meaning that the model will try to satisfy them at the cost of proper object detection. On the other hand, if $\lambda$ is too low, the constraints become increasingly "soft", meaning the neural network can violate them and may do it often to prioritize learning object detection. To strike a good balance between proper object detection and constraint violation is key to our approach.

We propose a scheduling mechanism that dynamically adjusts $\lambda$ during training. The scheduler follows an exponential function that starts at an initial value $\lambda_0$ that is increased to a multiple at every epoch. Note that this is in contrast to conventional learning rate schedulers, which usually *decrease* the learning rate over time. It allows the model to first learn the detection without receiving excessive influence, and to focus on constrained losses later in the training. Furthermore, it is beneficial to change $\lambda_0$ only after a warmup phase. This leads to the following Equation (3( for the $\lambda$ scheduler:

$$\lambda_t = \lambda_0 \ \text{if} \ t \leq t_w \ \text{else} \ \gamma \cdot \lambda_{t-1} \tag{3}$$

Here, $\lambda_t$ is the regularization at epoch $t$ and $t_w$ is the number of epochs we do not change the constrained loss; $\gamma \geq 1$ is the progression rate; for $\gamma = 1$, we have conventional (static) scheduling.

| $T$ | ROAD-R | | | | | | ROAD-Waymo-R | | | | | |
|---|---|---|---|---|---|---|---|---|---|---|---|---|
| | f-mAP@0.5 | | | MPBV % | | | f-mAP@0.5 | | | MPBV % | | |
| | 3DRN° | Yv8n | Y11n | 3DRN° | Yv8n | Y11n | I3D◇ | Yv8n | Y11n | I3D◇ | Yv8n | Y11n |
| $B$ | 33.39 | 57.55 | 61.43 | – | 40.45 | 39.52 | 16.38 | 34.88 | 34.33 | – | 47.85 | 46.04 |
| $T_\text{P}$ | 33.34 | **61.00** | 58.75 | – | 35.66 | 36.57 | – | 35.15 | 33.74 | – | 43.04 | 41.88 |
| $T_\text{G}$ | 32.16 | 50.23 | 55.87 | – | 41.75 | 35.70 | **17.10** | 32.54 | 30.33 | – | **34.17** | **40.48** |
| $T_\text{L}$ | **34.42** | 59.55 | **61.74** | – | 66.4 | 67.61 | 16.19 | 36.59 | 35.69 | – | 66.81 | 57.49 |
| $T_\text{HP}$ | – | 57.92 | 59.43 | – | 34.01 | 35.20 | – | 33.04 | 31.93 | – | 39.00 | 41.36 |
| $T_\text{NM}$ | – | 54.86 | 61.53 | – | 45.88 | 38.76 | – | 35.32 | 33.91 | – | 46.08 | 44.18 |
| $T_\text{D}$ | – | 50.48 | 60.08 | – | 50.55 | 36.11 | – | 34.33 | 33.21 | – | 46.74 | 42.36 |
| $T_\text{F}^2$ | – | 59.01 | 60.74 | – | 39.36 | 34.68 | – | 35.38 | 33.71 | – | 45.22 | 42.56 |
| $T_\text{Y}^2$ | – | 53.03 | 54.39 | – | **29.48** | **29.97** | – | 28.79 | 29.42 | – | 47.53 | 49.27 |
| $T_\text{SW}^1$ | – | 60.87 | 61.35 | – | 60.67 | 62.21 | – | **37.08** | 35.54 | – | 68.27 | 59.12 |
| $T_\text{AA}^1$ | – | 59.41 | 56.50 | – | 37.72 | 33.91 | – | 33.35 | 32.08 | – | 43.24 | 41.17 |
| $T_\text{H}^2$ | – | 59.21 | 59.59 | – | 36.46 | 40.32 | – | 36.10 | 34.13 | – | 49.27 | 44.65 |
| $T_\text{SS}^2$ | – | 59.95 | 60.04 | – | 61.32 | 57.90 | – | 35.57 | **35.89** | – | 63.96 | 49.78 |

Table 3: Combined accuracy (f-mAP@0.5) and constraint violation (MPBV %) results for individual t-norm training on ROAD-R ($\lambda = 50$) and ROAD-Waymo-R ($\lambda = 100$), across evaluation setups: 3D-RetinaNet, I3D (baselines), YOLOv8n or YOLO11n (ours) as vision models for *MOD-ECL*.

## 5 Methodology and Experiments

### 5.1 Methodology

Our methodology evaluates and optimizes t-norm performance on a given dataset by first assessing multiple t-norms. Using an adaptive algorithm, we automated t-norm selection during training and analyzed the frequency of use for each t-norm. Based on this analysis, we identified a subset of frequently selected t-norms and further evaluated their effectiveness through individual t-norm training, as shown in Table 2 (in our experiments, we included all t-norms). Subsequently, we selected the best-performing t-norm in terms of both detection accuracy and violation rate, and tuned the $\lambda$ parameter to examine its impact on detection performance and constraint satisfaction. Finally, a scheduler controlled by a parameter $\gamma$ was applied to optimize the tradeoff between detection accuracy and constraint satisfaction.

### 5.2 Experimental Setup

We conduct our experiments on the available data of ROAD-R and ROAD-Waymo-R. As the ground-truth labels for the test sets provided in ROAD and ROAD-Waymo are not public, we split the given videos into training and test sets (details are given in the repository). The computational resources used are specified in the Appendix. We use $MOD_\text{YOLO}$ on top of YOLOv8n and YOLO11n. We first produce a baseline without constrained loss, and then with each t-norm mentioned in Section 4 individually. We then present results for the adaptive t-norm algorithm varying the parameters $\beta$ and $\delta$, which represent the probability of random exploration, and the discount factor respectively. We conclude by experimenting on $\lambda$ by using extreme values and a scheduler for $\lambda$ that increases its value at every epoch.

All results were calculated with frame-mAP with IOU 0.5 (f-mAP@0.5) as in [15, 12]. The mAP for each score was calculated using the torchmetrics library. Furthermore, we calculated the percentage of violated bounding boxes (sets of labels that violate at least one constraint), which we refer to as *Mean Per Box Violation (MPBV)*. The best results are shown in bold and the worst in gray.

### 5.3 Experimental Results

**Individual T-Norms.** The results for our baseline ($B$) and individual t-norms in Table 2 are presented in Table 3. The baseline does not use constrained loss, i.e., $\lambda = 0$. For the experiments with t-norms, we used $\lambda = 50, 100$ for ROAD-R, ROAD-Waymo-R respectively as it provides favorable tradeoffs. For comparison, we present the state-of-the-art results given in [26, 21] for ROAD-R (3D-RetinaNet) and ROAD-Waymo-R (I3D) respectively, marked with superscript ∘ and ⋄.

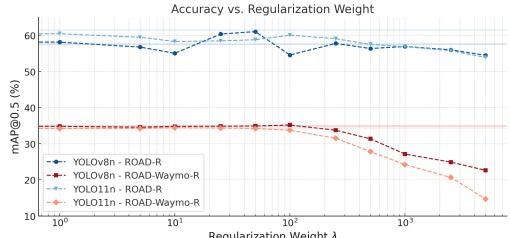
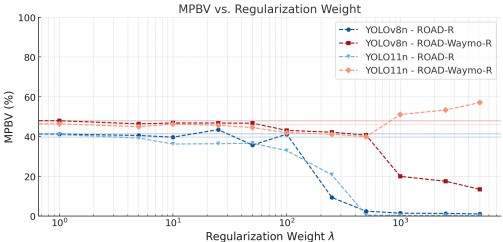

(a) Accuracy (mAP@0.5) as a function of regularization weight $\lambda$ (log-scale).

(b) Mean Per Bounding Box Violation (MPBV) as a function of regularization weight $\lambda$ (log-scale).

Figure 3: Comparison of accuracy and violation rate across YOLOv8n and YOLO11n models on ROAD-R and ROAD-Waymo-R.

| Params $(\beta, \delta)$ | ROAD-R | | | | ROAD-Waymo-R | | | |
| --- | --- | --- | --- | --- | --- | --- | --- | --- |
| | f-mAP@0.5 | | MPBV (%) | | f-mAP@0.5 | | MPBV (%) | |
| | Yv8n | Y11n | Yv8n | Y11n | Yv8n | Y11n | Yv8n | Y11n |
| $B$ | 57.55 | **61.43** | 40.45 | 39.52 | **34.88** | **34.33** | 47.85 | 46.04 |
| (0.1, 0.25) | 55.10 | 56.94 | **30.20** | 34.95 | 33.27 | 30.57 | 44.24 | 43.41 |
| (0.1, 0.5) | 53.88 | 56.99 | 31.75 | 37.21 | 33.67 | 31.37 | 44.27 | 45.58 |
| (0.25, 0.25) | 57.19 | 56.93 | 31.52 | 39.39 | 33.15 | 31.13 | 42.78 | **42.13** |
| (0.25, 0.5) | 52.28 | 58.25 | 32.28 | 35.00 | 33.29 | 31.40 | 43.15 | 42.47 |
| (0.5, 0.5) | **58.93** | 57.37 | 36.34 | **34.85** | 33.72 | 32.28 | **41.75** | 42.65 |

| Params $\gamma$ | ROAD-R | | | | ROAD-Waymo-R | | | |
| --- | --- | --- | --- | --- | --- | --- | --- | --- |
| | f-mAP@0.5 | | MPBV (%) | | f-mAP@0.5 | | MPBV (%) | |
| | Yv8n | Y11n | Yv8n | Y11n | Yv8n | Y11n | Yv8n | Y11n |
| $B$ | 57.55 | **61.43** | 40.45 | 39.52 | 34.88 | **34.33** | 47.85 | 46.04 |
| 1.1 | **60.16** | 58.77 | 38.73 | 35.66 | **35.22** | 32.35 | 46.19 | 36.26 |
| 1.25 | 57.18 | 57.74 | 10.36 | 19.48 | 35.04 | 32.84 | 42.88 | **31.40** |
| 1.5 | 58.60 | 56.17 | 10.53 | 0.019 | 34.27 | 25.56 | 37.52 | 32.41 |
| 1.75 | 55.00 | 55.67 | 0.698 | 0.006 | 32.64 | 28.70 | **18.98** | 35.41 |
| 2.0 | 55.18 | 55.04 | 0.722 | **0.005** | 31.88 | 21.66 | 45.13 | 43.66 |

(a) Adaptive algorithm using $\lambda = (50, 100)$ with parameters $(\beta, \delta)$.

(b) $\lambda$ scheduler effect using $\lambda_0 = (50, 100)$ and varying $\gamma$.

Table 4: Adaptive algorithm and $\lambda$-scheduler performance across ROAD-R and ROAD-Waymo-R.

**Adaptive Algorithm.** The results for our adaptive algorithm are in Table 4a for both datasets. We experimented on $T_\mathrm{P}$ with exploration values $\beta = \{0.1, 0.25, 0.5\}$ and discount constant $\delta = \{0.25, 0.5\}$. The percentual frequency each t-norm was used is presented in Figure 4. $\lambda = (50, 100)$ was used for ROAD-R and ROAD-Waymo-R respectively.

**Varying** $\lambda$. We select just one t-norm to reduce the number of experiments, namely $T_\mathrm{P}$, with $\lambda = \{0, 1, 5, 10, 50, 100, 500, 1000, 5000\}$. Results are shown in Figure 3.

$\lambda$ **Scheduler.** The results of using our scheduler are shown in Table 4b. We used $T_\mathrm{P}$ with an initial $\lambda_0 = (50, 100)$ for ROAD-R and ROAD-Waymo-R respectively, with warm-up $t_w = 3$ and $\gamma = \{1.1, 1.25, 1.5, 1.75, 2.0\}$.

## 5.4 Discussion of Results

Our results show that the integration of logical constraints through constrained loss leads to significant improvements in requirement satisfaction, while in many cases maintaining or even enhancing detection performance. Looking at Table 3, several t-norms, including $T_\mathrm{SW}^1, T_\mathrm{SS}^2$ and $T_\mathrm{L}$ in most cases achieved higher f-mAP@0.5 than the baseline without constrained loss, suggesting that enforcing logical consistency does not necessarily hinder object detection performance. Focusing on the t-norms $T_\mathrm{P}, T_\mathrm{G}$ and $T_\mathrm{L}$, we can see an interesting relationship between them. $T_\mathrm{P}$ presents the most balanced behavior overall, consistently achieving good tradeoffs for accuracy and reducing violations. $T_\mathrm{G}$ focuses on reducing violations over detection performance. Last, $T_\mathrm{L}$ works conversely by gaining accuracy at the cost of more violations. Interestingly, $T_\mathrm{Y}^2$ reduces the violations by the most in the case of ROAD-R. We believe this is because the formula allows the calculation of very low values of violations. For ROAD-Waymo-R this is not the case, where it actually hinders both accuracy and compliance. This shows the difficulties of deciding which t-norms work in a specific setting, indicating the need for a comprehensive framework such as ours for evaluating each t-norm.

Beyond static t-norm evaluation, the adaptive algorithm reveals t-norm preferences across training configurations. Table 4a presents the results for our adaptive algorithm using $T_\mathrm{P}$. For all combinations of $(\beta, \delta)$, we get lower violations than the baseline, which was intended by the decision of using the constrained loss as the score. While this comes at the consequence of decreased accuracy, the

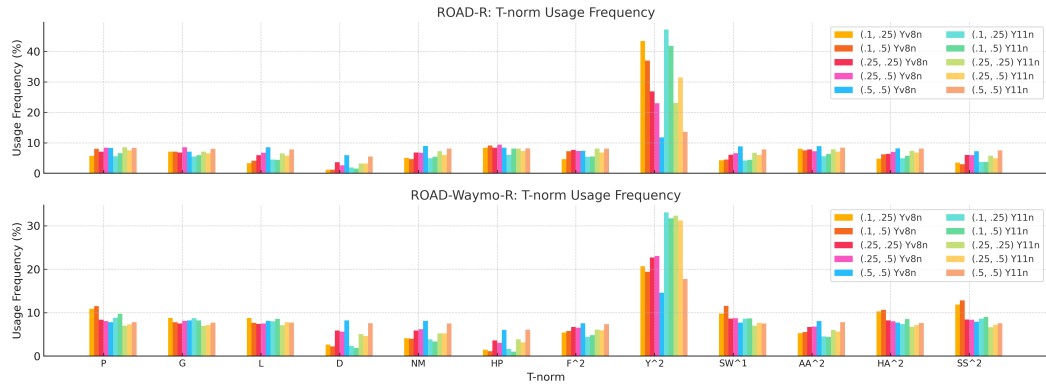

Figure 4: Frequency of t-norm selection by the adaptive algorithm under different $(\beta, \delta)$ configurations. Top: ROAD-R dataset; Bottom: ROAD-Waymo-R dataset.

reduction in violation is usually higher than the accuracy reduction, depending on the configuration. Using some configurations may reduce the violations more at the cost of more accuracy, revealing a tradeoff when using the adaptive algorithm. Figure 4 visualizes the frequency of each t-norm's selection under varying $(\beta, \delta)$ values. On ROAD-R and ROAD-Waymo-R, $T_Y^2$ is dominant, especially under conservative exploration settings (e.g., $\beta = 0.1$, $\delta = 0.25$), being selected over 40% of the time. This is supported by previous results using it in a fixed manner. This makes sense for ROAD-R but not for ROAD-Waymo-R, where we see $T_Y^2$ having a detrimental effect on constraint satisfaction in the static case. These results indicate that while in the static case, some t-norms appear better than others, it is not the case throughout the entire training process. Different t-norms may work better at different stages of the training, shown by having a better tradeoff between violation and accuracy than $T_Y^2$ in the static case.

Our study of the regularization term $\lambda$ reveals that its effect is non-linear, shown in Figure 3. As $\lambda$ increases, violation rates decrease significantly, but accuracy begins to plateau or decline beyond certain thresholds. For instance, increasing $\lambda$ from 50 to 500 reduces MPBV but leads to diminishing accuracy gains. This confirms that excessive emphasis on constraints may limit detection capability, and that $\lambda$ must be tuned to the desired balance. To address this, we proposed a scheduler that gradually increases $\lambda$ during training. The results show that moderate growth rates (e.g., $\gamma = 1.25$) reduce violations while preserving accuracy, supporting the hypothesis that applying constrained loss is most beneficial in later stages of training. Table 4b shows that for every configuration, the violations are always lower than the baseline at a low cost of accuracy. Starting from $\gamma = 1.1$ which has the mildest effect, one can increase the value to trade accuracy for compliance. This is expected as higher values of $\lambda$ magnify the effect of the constrained loss. However, we should be careful of excessively increase it, as in cases such as ROAD-Waymo-R, the compliance worsens with higher values, e.g., $\gamma = 2$.

## 6 Conclusion and Future Work

We presented the *MOD-ECL* framework for object detection in AD that enforces logical constraints using t-norms, along with novel algorithms for adaptive t-norm selection and constrained loss regularization. The results for challenging AD datasets show that this approach can significantly reduce logical violations as well as enhance detection performance. Furthermore, adaptive t-norm selection and dynamic regularization optimizes the balance between detection and constraint satisfaction.

Future work will explore the potential of leveraging more powerful models within the YOLO family and longer training periods for enhanced detection performance. Additionally, investigating different scoring mechanisms for the adaptive algorithm and experimenting with alternative functions for the $\lambda$ scheduler could further refine the balance between detection accuracy and requirement adherence. Finally, deployed on *MOD-ECL*, our methodology enabled a structured exploration and effective application of t-norms in AD with constraints. While we did not consider other scenarios (e.g., healthcare), we suspect that this approach is also beneficial in them, which remains to be explored.

## Acknowledgments

This work has been supported by JSPS KAKENHI Grant Number JP25K03190, JST CREST Grant Number JPMJCR22D3 and JST SPRING Grant Number JPMJSP2104, Japan. This research was funded in whole or in part by the Austrian Science Fund (FWF) 10.55776/COE12. We also thank the JASEC-NII collaboration program.

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

# A    Appendix

This appendix contains supplementary material not present in the main body of the document. It includes a link containing the logs of our experiments, instructions on how to setup and reproduce our results, additional results and implementation details of the t-norms used.

## A.1    Evaluation platform

We used two computational nodes for the experiments to handle the large number of experiments: one node, used for ROAD-R, had 4 NVIDIA RTX A5000 GPUs, 2 Intel Xeon Silver 4314 CPU, and a total of 1024GB RAM; the other node, used for ROAD-Waymo-R, was equipped with 2 NVIDIA RTX A6000 GPUs, 1 AMD Ryzen Threadripper PRO 5995WX CPU with 64 cores, and a total of 256GB RAM.

## A.2    Experimental Logs

All the resulting logs of our experiments can be found via the following link: `https://drive.google.com/file/d/1P7-mgNrIcJbQg5ZAHSfrj1w8W-1AnskI/view?usp=sharing`

## A.3    Setup and Experimental Reproduction

The instructions can be found in the repository of our submission repository: `https://anonymous.4open.science/r/neurips25`

# B    Additional Tables and Graphs

In this section, we provide additional qualitative results showing how the use of constrained loss influences both the accuracy of predictions and the satisfaction of logical constraints. These visualizations support the findings discussed in Sections 5, 5.4, illustrating how the combination of t-norm selection and $\lambda$ directly impacts both the model's object detection accuracy and its compliance to logical constraints.

Table 5 contains the values that were graphed in Figure 3. As $\lambda$ increases, models better satisfy constraints (lower MPBV) but at the cost of detection accuracy (f-mAP@0.5). YOLOv8n shows a smoother degradation than YOLO11n. A balance point occurs around $\lambda = 25$–$100$, after which accuracy suffers significantly. Table 6 shows the percentual frequencies that were graphed in Figure 4. Across both ROAD-R and ROAD-Waymo-R, the adaptive algorithm favors the Yager t-norm ($T_Y^2$), consistently selecting it more than any other. In contrast, $T_D$ and $T_{HP}$ are rarely chosen for ROAD-R and ROAD-Waymo-R respectively, suggesting they are generally less effective for their constraint-driven training dataset.

The expanded t-norm set and adaptive algorithm may introduce additional computational costs. To quantify these, we provide a comparative breakdown of GPU memory usage, FLOPs, total training time, and per-epoch wall-clock time relative to the baseline and previously published constrained-loss methods. Our method incurs no additional overhead at inference time, as t-norms are used exclusively during training. Table 7 summarizes mean values (± standard deviation) and 95% confidence intervals (CI) for mAP and MPBV over three seeds (0, 1, 2) for each configuration. Exceptions are adaptive YOLOv8n and YOLO11n on ROAD-Waymo-R, where the former was run with seed 0 only and the latter with seeds 0 and 1. Also reported are total training time, FLOPs, and peak GPU memory usage. It should be noted that there are no additional costs incurred at inference time.

The following figures graph mAP@0.5 against MPBV. Here it easy to divide the figure into four subquadrants defined by the position of the baseline: (1) top-left represents the experiments that increased accuracy and reduced violations (best case), (2) top-right represents the experiments that increased accuracy and violations, (3) bottom-left represents the ones that decreased accuracy and violations (trade-off cases), and (4) bottom-right those that decreased accuracy and increased violations (worst case). Figure 5 shows graphs of all model and dataset combinations using static t-norms. Please observe that the faded triangles connecting the t-norms $T_P$, $T_G$ and $T_L$ are in similar positions for each model and dataset, as mentioned in Section 5.4 of the main text. Figures 6 and 7 show every configuration of the adaptive algorithm and the $\lambda$ scheduler, respectively. The faded line

that goes through the baseline has slope 1. All points to the left of the line have a positive trade-off, i.e., we get a bigger improvement in violation compared to f-mAP@0.5.

| | ROAD-R | | | | ROAD-Waymo-R | | | |
|---|---|---|---|---|---|---|---|---|
| | f-mAP@0.5 | | MPBV (%) | | f-mAP@0.5 | | MPBV (%) | |
| $\lambda$ | Yv8n | Y11n | Yv8n | Y11n | Yv8n | Y11n | Yv8n | Y11n |
| 0 | 57.55 | 61.43 | 41.16 | 39.52 | 34.88 | 34.30 | 47.85 | 46.04 |
| 1 | 58.09 | 60.40 | 41.16 | 40.92 | 34.80 | 34.20 | 47.89 | 46.31 |
| 5 | 56.72 | 59.45 | 40.40 | 39.17 | 34.54 | 34.20 | 46.33 | 44.89 |
| 10 | 55.01 | 58.26 | 39.63 | 36.18 | 34.78 | 34.40 | 46.77 | 46.23 |
| 25 | 60.33 | 58.44 | 43.40 | 37.28 | 34.82 | 34.33 | 46.72 | 45.36 |
| 50 | 61.00 | 58.75 | 35.66 | 36.57 | 34.89 | 34.20 | 46.66 | 44.48 |
| 100 | 54.53 | 60.02 | 41.12 | 32.85 | 35.15 | 33.70 | 43.04 | 41.88 |
| 250 | 57.74 | 59.05 | 9.25 | 16.61 | 33.72 | 31.49 | 41.83 | 40.86 |
| 500 | 56.32 | 57.44 | 2.39 | 0.38 | 31.33 | 27.80 | 40.62 | 39.84 |
| 1000 | 56.89 | 56.84 | 1.38 | 0.24 | 27.11 | 24.20 | 19.89 | 51.03 |
| 2500 | 55.97 | 55.72 | 1.19 | 0.21 | 24.89 | 20.64 | 16.64 | 54.04 |
| 5000 | 54.45 | 53.85 | 1.00 | 0.18 | 22.62 | 14.70 | 13.38 | 57.04 |

Table 5: Comparison of detection accuracy (f-mAP@0.5) and constraint violation (MPBV) for YOLOv8n and YOLO11n across ROAD-R and ROAD-Waymo-R, with different $\lambda$ values and $T_{\mathrm{P}}$ t-norm.

| Params $(\beta, \delta)$ | $T_{\mathrm{P}}$ | | $T_{\mathrm{G}}$ | | $T_{\mathrm{L}}$ | | $T_{\mathrm{D}}$ | | $T_{\mathrm{NM}}$ | | $T_{\mathrm{HP}}$ | | $T_{\mathrm{F}}^2$ | | $T_{\mathrm{Y}}^2$ | | $T_{\mathrm{SW}}^1$ | | $T_{\mathrm{AA}}^2$ | | $T_{\mathrm{H}}^2$ | | $T_{\mathrm{SS}}^2$ | |
|---|---|---|---|---|---|---|---|---|---|---|---|---|---|---|---|---|---|---|---|---|---|---|---|---|
| | Yv8n | Y11n | Yv8n | Y11n | Yv8n | Y11n | Yv8n | Y11n | Yv8n | Y11n | Yv8n | Y11n | Yv8n | Y11n | Yv8n | Y11n | Yv8n | Y11n | Yv8n | Y11n | Yv8n | Y11n | Yv8n | Y11n |
| (.1, .25) | 5.7 | 5.6 | 7.1 | 5.5 | 3.3 | 4.5 | 1.2 | 1.9 | 5.1 | 4.9 | 8.4 | 6.0 | 4.6 | 5.4 | **43.4** | **47.2** | 4.3 | 4.2 | 8.0 | 5.6 | 4.8 | 4.9 | 3.5 | 3.7 |
| (.1, .5) | 8.0 | 6.6 | 7.1 | 5.9 | 4.1 | 4.4 | 1.2 | 1.5 | 4.6 | 5.4 | 9.1 | 8.1 | 7.2 | 5.5 | **37.0** | **41.8** | 4.5 | 4.4 | 7.5 | 6.4 | 6.2 | 5.7 | 3.0 | 3.7 |
| (.25, .25) | 7.1 | 8.5 | 6.8 | 7.1 | 5.9 | 6.5 | 3.6 | 3.2 | 6.8 | 7.2 | 8.4 | 8.1 | 7.7 | 8.1 | **26.9** | **23.1** | 6.1 | 6.7 | 7.8 | 7.9 | 6.4 | 7.4 | 6.0 | 5.8 |
| (.25, .5) | 8.4 | 7.5 | 8.5 | 6.6 | 6.7 | 5.8 | 2.6 | 3.2 | 6.7 | 6.0 | 9.4 | 7.3 | 7.3 | 6.8 | **23.0** | **31.4** | 6.6 | 6.0 | 7.2 | 7.1 | 7.0 | 6.8 | 5.9 | 4.9 |
| (.5, .5) | 8.3 | 8.3 | 7.1 | 8.0 | 8.5 | 7.8 | 5.9 | 5.5 | 9.0 | 8.1 | 8.4 | 8.2 | 7.4 | 8.1 | **11.8** | **13.6** | 8.8 | 7.8 | 8.9 | 8.4 | 8.2 | 8.1 | 7.2 | 7.5 |
| (.1, .25) | 10.9 | 8.8 | 8.8 | 8.7 | 8.8 | 8.0 | 2.7 | 2.3 | 4.1 | 3.9 | 1.5 | 1.5 | 5.5 | 4.4 | **20.7** | **33.1** | 9.8 | 8.6 | 5.3 | 4.5 | 10.3 | 7.3 | 11.9 | 8.6 |
| (.1, .5) | 11.5 | 9.8 | 7.8 | 8.2 | 7.6 | 8.6 | 2.2 | 1.9 | 4.0 | 3.4 | 1.1 | 1.0 | 5.8 | 4.8 | 11.6 | **19.4** | **31.7** | 8.7 | 5.5 | 4.4 | 10.6 | 8.5 | 12.8 | 9.0 |
| (.25, .25) | 8.3 | 7.0 | 7.5 | 6.9 | 7.4 | 7.1 | 5.9 | 5.0 | 5.9 | 5.2 | 3.6 | 3.9 | 6.7 | 6.1 | **22.7** | **32.3** | 8.6 | 7.0 | 6.7 | 6.0 | 8.2 | 6.8 | 8.4 | 6.7 |
| (.25, .5) | 8.1 | 7.3 | 8.1 | 7.2 | 7.5 | 7.8 | 5.6 | 4.6 | 6.2 | 5.2 | 3.1 | 3.2 | 6.5 | 5.9 | **23.0** | **31.2** | 8.7 | 7.6 | 6.8 | 5.6 | 8.0 | 7.1 | 8.4 | 7.2 |
| (.5, .5) | 7.8 | 7.8 | 8.2 | 7.7 | 8.1 | 7.7 | 8.2 | 7.6 | 8.1 | 7.5 | 6.0 | 6.1 | 7.6 | 7.4 | **14.6** | **17.7** | 7.7 | 7.5 | 8.1 | 7.8 | 7.7 | 7.6 | 7.9 | 7.6 |

Table 6: Adaptive algorithm t-norm usage frequency (%) for YOLOv8n (Yv8n) and YOLO11n (Y11n) across different $\beta, \delta$ parameter settings in ROAD-R (top) and ROAD-Waymo-R (bottom). Most used values bolded; least grayed.

| Model | Config | mAP | MPBV | Train Time (s) | FLOPs (G) | Peak Mem (GB) |
|---|---|---|---|---|---|---|
| v8n | Baseline | 56.34 ± 0.6 (CI 1.98) | 41.18 ± 1.3 (CI 2.0) | 1176 ± 31.96 | 4.12 | 71.38 ± 3.28 |
| | Product ($\lambda = 50$) | 60.77 ± 1.0 (CI 2.61) | 36.56 ± 1.2 (CI 1.9) | 2027 ± 30.72 | 4.12 | 73.25 ± 0 |
| | Adaptive ($\beta = 0.5, \gamma = 0.5$) | 59.02 ± 0.8 (CI 2.54) | 36.29 ± 0.9 (CI 1.5) | 2800 ± 87.93 | 4.12 | 73.25 ± 0 |
| 11n | Baseline | 61.69 ± 0.8 (CI 2.03) | 40.18 ± 3.4 (CI 6.6) | 1481 ± 32.58 | 3.24 | 77.24 ± 0 |
| | Product ($\lambda = 50$) | 58.06 ± 1.0 (CI 2.50) | 36.38 ± 2.2 (CI 5.1) | 2319 ± 24.27 | 3.24 | 77.24 ± 0 |
| | Adaptive ($\beta = 0.5, \gamma = 0.5$) | 57.94 ± 0.9 (CI 2.42) | 34.67 ± 1.4 (CI 3.6) | 3374 ± 26.89 | 3.24 | 77.24 ± 0 |
| v8n | Baseline | 34.96 ± 0.27 (CI 0.78) | 47.19 ± 0.88 (CI 2.48) | 15874 ± 70.35 | 4.12 | 23.04 ± 0 |
| | Product ($\lambda = 100$) | 35.12 ± 0.05 (CI 0.14) | 43.63 ± 0.81 (CI 2.30) | 21258 ± 233.18 | 4.12 | 23.04 ± 0 |
| | Adaptive ($\beta = 0.5, \gamma = 0.5$) | 33.51 ± 0.00 (CI 0.00) | 45.59 ± 0.00 (CI 0.00) | 24903 ± 0.00 | 4.12 | 23.04 ± 0 |
| 11n | Baseline | 34.65 ± 0.28 (CI 0.79) | 46.32 ± 0.40 (CI 1.12) | 16254 ± 33.47 | 3.25 | 23.44 ± 0 |
| | Product ($\lambda = 100$) | 33.64 ± 0.08 (CI 0.24) | 42.99 ± 1.01 (CI 2.84) | 20157 ± 74.87 | 3.25 | 23.44 ± 0 |
| | Adaptive ($\beta = 0.5, \gamma = 0.5$) | 32.91 ± 0.42 (CI 1.17) | 43.02 ± 0.53 (CI 1.50) | 25983 ± 1356.45 | 3.25 | 23.44 ± 0 |

Table 7: Computational overhead comparison of static and adaptive t-norm selection. Top: ROAD-R results. Bottom: ROAD-Waymo-R results. Mean ± standard deviation and 95% confidence intervals (CI) are shown for mAP and MPBV over three seeds unless otherwise noted. No additional inference-time cost is incurred.

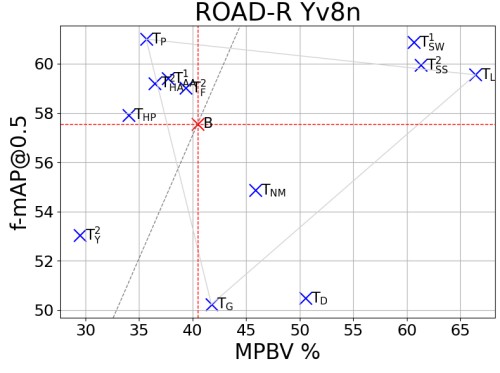

(a) Accuracy against MPBV on ROAD-R using YOLOv8n.

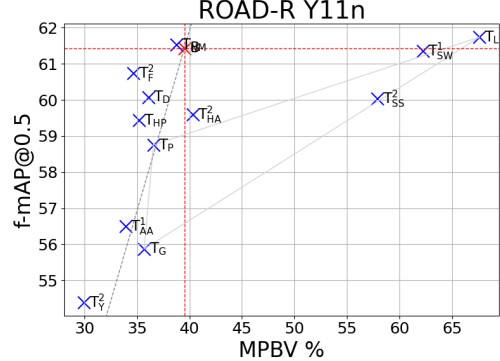

(b) Accuracy against MPBV on ROAD-R using YOLO11n.

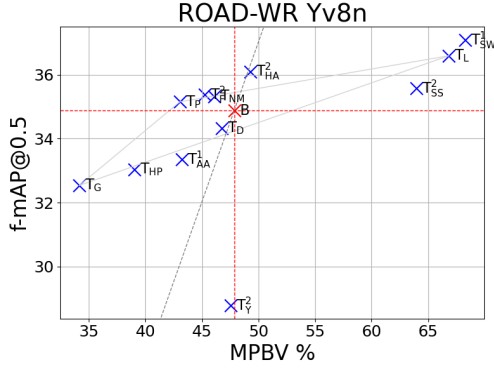

(c) Accuracy against MPBV on ROAD-Waymo-R using YOLOv8n.

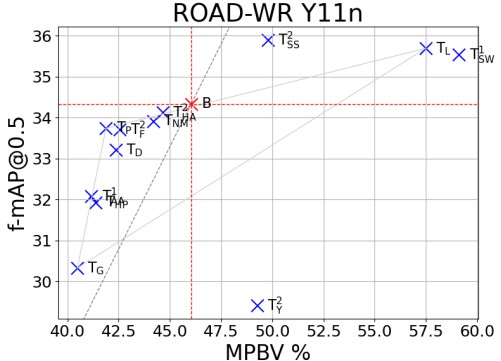

(d) Accuracy against MPBV on ROAD-Waymo-R using YOLO11n.

Figure 5: Evaluation of static t-norm comparisons across datasets. The faded triangle connects $T_P$, $T_G$ and $T_L$.

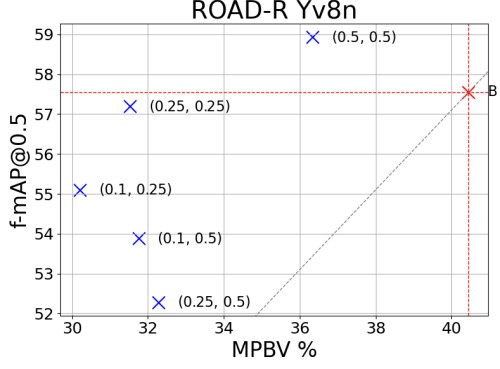

(a) Accuracy against MPBV on ROAD-R using YOLOv8n.

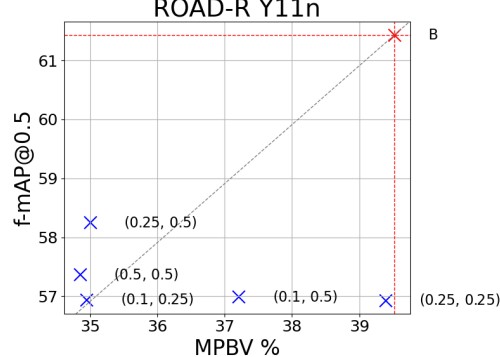

(b) Accuracy against MPBV on ROAD-R using YOLO11n.

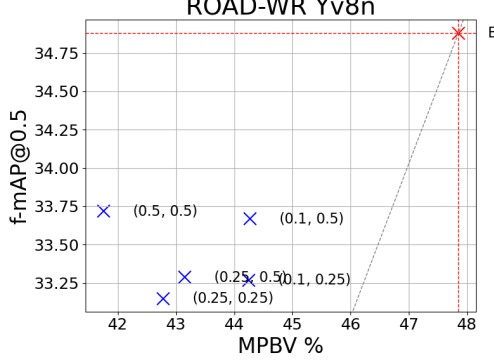

(c) Accuracy against MPBV on ROAD-Waymo-R using YOLOv8n.

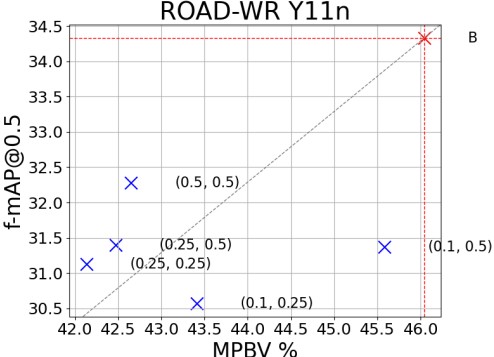

(d) Accuracy against MPBV on ROAD-Waymo-R using YOLO11n.

Figure 6: Evaluation of the adaptive algorithm across datasets. The faded line that goes through the baseline has slope 1.

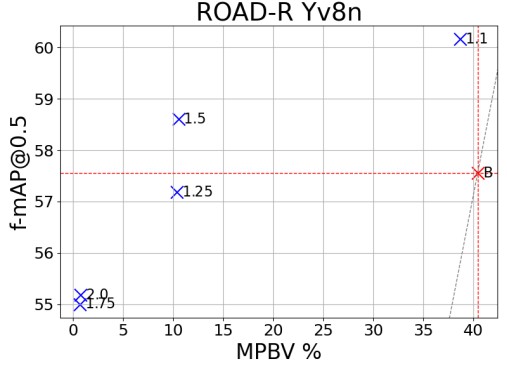

(a) Accuracy against MPBV on ROAD-R using YOLOv8n.

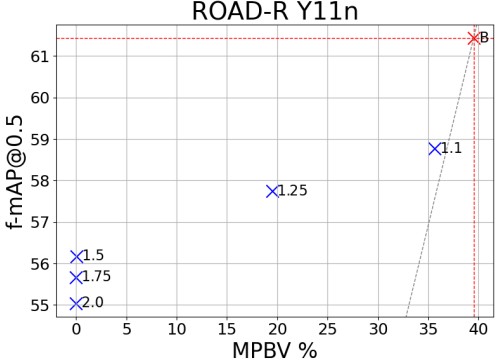

(b) Accuracy against MPBV on ROAD-R using YOLO11n.

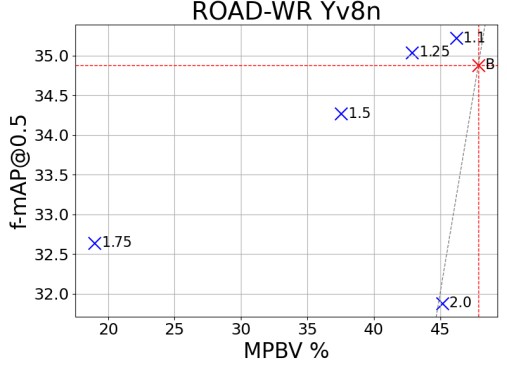

(c) Accuracy against MPBV on ROAD-Waymo-R using YOLOv8n.

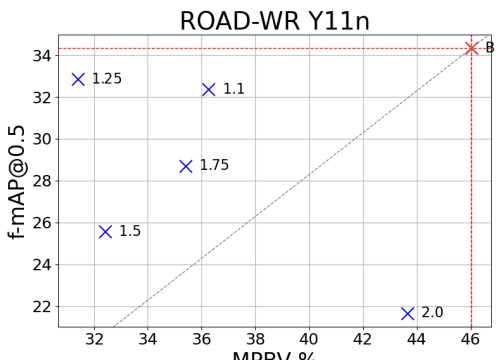

(d) Accuracy against MPBV on ROAD-Waymo-R using YOLO11n.

Figure 7: Evaluation of the $\lambda$ scheduler across datasets. The faded line that goes through the baseline has slope 1.

## C   Theoretical Guarantees

**Theorem 1.** *Let $N_T(n)$ denote the number of times $T$ is selected up to iteration $n$. For any suboptimal t-norm $T$ whose expected discounted normalized gain $\mu_T$ is strictly lower than that of the best t-norm $T^*$ (i.e., $\mu_T < \mu_{T^*}$), the expected number of times $T$ is selected satisfies the following upper bound:*

$$\mathbb{E}[N_T(n)] \leq \frac{\log n}{\beta \cdot \Delta_T^2}$$

*where $\Delta_T := \mu_{T^*} - \mu_T$ is the expected discounted performance gap.*

*Proof.* Let $T^*$ be the optimal t-norm with the highest expected normalized gain:

$$\mu_T := \mathbb{E}[n_i \mid T_i = T] \quad \text{and} \quad \mu_{T^*} := \max_{T \in \mathcal{T}} \mu_T$$

Let $T$ be any suboptimal t-norm such that $\Delta_T := \mu_{T^*} - \mu_T > 0$.

We consider the structure of the score update:

$$\texttt{scores}[T] \leftarrow \delta \cdot \texttt{scores}[T] + (1 - \delta) \cdot n_i$$

This is a form of discounted average which places more weight on recent normalized improvements. Define:

$$s_t^T := \sum_{j=1}^{t} (1 - \delta)\delta^{t-j} \cdot n_j^T$$

as the effective score of $T$ after $t$ updates, where $n_j^T$ is the normalized gain at time $j$ when $T$ was chosen.

Since $n_i^T$ are bounded (due to normalization by $\epsilon$ or $cl_p$), and the gain distributions are stationary (or slowly drifting), standard concentration inequalities (e.g., Azuma-Hoeffding for martingale differences) imply that with high probability, $s_t^{T^*}$ will dominate $s_t^T$ after a sufficient number of iterations.

Because greedy selections use $\arg\max_T \texttt{scores}[T]$, the probability of selecting a suboptimal $T$ due to exploitation decays exponentially over time. However, exploration ensures that $T$ is still selected occasionally.

Thus, the algorithm favors the optimal $T^*$ in the long run while bounding the frequency of suboptimal t-norm use, concluding the proof. □

**Theorem 2.** *Let $\mathcal{T}$ be a finite set of t-norms. Suppose the constraint loss for each $T \in \mathcal{T}$ varies stochastically over time and let $n_i$ denote the normalized improvement at iteration $i$ as computed by Algorithm 1. Then, under a fixed exploration rate $\beta > 0$ and discount factor $\delta \in (0, 1)$, the adaptive t-norm selection algorithm ensures that each $T \in \mathcal{T}$ is selected infinitely often with probability 1 as the number of iterations $n \to \infty$.*

*Proof.* At each iteration $i$, the algorithm chooses a t-norm $T_i$ either randomly with probability $\beta > 0$ or greedily according to the highest current score. Because $\mathcal{T}$ is finite and $\beta$ is constant, each $T \in \mathcal{T}$ has a non-zero probability of being selected at each iteration, where $K = |\mathcal{T}|$:

$$\mathbb{P}(T_i = T) \geq \frac{\beta}{K}$$

Thus, by the Borel–Cantelli lemma and standard results in stochastic processes, each $T$ is sampled infinitely often with probability 1 as $n \to \infty$. This ensures that each score estimate receives updates infinitely often.

□

## D Implementation of t-norms using `PyTorch`

We make use of the `PyTorch` python module to efficiently implement t-norms processed in batches. For a tensor $\mathbf{T}$ of shape $(d_1, d_2, \ldots, d_n)$, slicing it column-wise along the last dimension:

$$\mathbf{T}_{:,:,\ldots,i}, \quad i \in \{0, \ldots, d_n - 1\}$$

This selects all elements across the first $n-1$ dimensions for a fixed $i$ in the last dimension. Consider all input values are `PyTorch` tensors. $fv = \mathbf{T}$ is the full set of bounding boxes predicted and scores associated to the labels. Otherwise, we slice the input tensor and operate iteratively, which is the case when we have $a, b$ as input, representing tensor slices to be operated with the iterative algorithm described on Section 3.

Listing 1: Minimum t-norm

```python
def min_tnorm_tensor(fv):
    min_value, _ = torch.min(fv, axis=-1)
    return min_value
```

Listing 2: Product t-norm

```python
def product_tnorm_tensor(fv):
    product_value = torch.prod(fv, axis=-1)
    return product_value
```

Listing 3: Lukasiewicz t-norm

```python
def lukasiewicz_tnorm_tensor(fv):
    sum_value = torch.sum(fv, axis=-1)
    return torch.relu(sum_value - fv.size(-1) + 1)
```

Listing 4: Drastic t-norm

```python
def drastic_tnorm_batch(a, b):
    none_one = (a == 1) | (b == 1)
    drastic_value = torch.where(none_one, torch.min(a, b), torch.
        zeros_like(a))
    return drastic_value
```

Listing 5: Nilpotent minimum t-norm

```python
def nilpotentmin_tnorm_batch(a, b):
    nilpotent_value = torch.where(a + b > 1, torch.min(a, b), torch.
        zeros_like(a))
    return nilpotent_value
```

Listing 6: Hamacher product t-norm

```python
def hamacherprod_tnorm_batch(a, b):
    denominator = torch.clamp(a + b - a * b, min=1e-6, max=1)
    hamacher_value = torch.where(a + b == 0, torch.zeros_like(a), (a *
        b) / denominator)
    return hamacher_value
```

Listing 7: Yager t-norm

```python
def yager_tnorm_batch(a, b, p=2):
    if p == 1:
        return lukasiewicz_tnorm(a, b)

    yager_value = torch.relu(1 - ((1 - a) ** p + (1 - b) ** p)) ** (1
        / p)
    return yager_value
```

Listing 8: Frank t-norm

```python
def frank_tnorm(a, b, p=2):
    if p == 1:
        return product_tnorm(a, b)

    numerator = (a**p - 1) * (b**p - 1)
    denominator = p - 1
    frank_value = log_with_base(1 + numerator / denominator, p)
    return frank_value
```

Listing 9: Sugeno-Weber t-norm

```python
def sugeno_weber_tnorm(a, b, p=1):
    sugeno_value = torch.max(torch.zeros_like(a), (a + b - 1 + p * a *
        b) / (1 + p))
    return sugeno_value
```

Listing 10: Aczel-Alsina t-norm

```python
def aczel_alsina_tnorm_batch(a, b, p=2):
    if p == 0:
        return drastic_tnorm_batch(a, b)

    none_zero = (a >= 1e-6) & (b >= 1e-6)
    a, b = torch.clamp(a, min=1e-6), torch.clamp(b, min=1e-6)

    exponent = ((torch.abs(-torch.log(a)) ** p + torch.abs(-torch.log(
        b)) ** p) ** (1 / p))
    aczel_value = torch.where(none_zero, torch.exp(-exponent), torch.
        zeros_like(a))
    return aczel_value
```

Listing 11: Hamacher t-norm

```python
def hamacher_tnorm(a, b, p=2):
    denominator = p + (1 - p) * (a + b - a * b)
    hamacher_value = a * b / denominator
    return hamacher_value
```

Listing 12: Schweizer-Sklar t-norm

```python
def schweizer_sklar_tnorm(a, b, p):
    inner_value = a**p + b**p - 1
    tnorm_value = torch.relu(inner_value) ** (1 / p)
    return tnorm_value
```

