# OpenReview forum: "T-norm Selection for Object Detection in Autonomous Driving with Logical Constraints"
_NeurIPS.cc/2025/Conference — NeurIPS 2025 poster_

### Official Review · Reviewer_DvBY · 2025-06-30

**Clarity:** 3
**Significance:** 2
**Originality:** 2
**Rating:** 4
**Confidence:** 3

**Summary:**

The main contributions of the paper are as follows:
1. T-norm selection: The authors device an algorithm which dynamically selects the best t-norm based on a score. This t-norm selection lets the model explore different t-norms during training and the score helps the user understand why a certain t-norm was selected.
2. Dynamic scheduler: The model learns to satisfy the t-norms by adding a constrained loss term on top of the object detection loss. To help the model balance both losses, the authors add a scheduler which weighs the constrained loss. Based on the scheduler, the authors hope that the model will learn to do object detection in the early stages of training and then focus on satisfying the t-norms by increasing the weight of the constrained loss term in the later parts of training.
3. MOD-ECL: A model based on MOD-YOLO, with the addition of adaptive t-norm selection and dynamically scheduled constrained loss to optimize model performance.

**Questions:**

1. In section 3.1, it would be good to give some examples for (o, l, a).
2. Equation 2 would be easier to understand if it is supported by real examples of constrains and t-norms.
3. What is the significance of the multi-label part of MOD-ECL? Does it need to be multi-label?
4. Why not use multiple t-norms for inference?
5. Merging section 6 into 5.3 might be more intuitive to the reader.

**Ethical Concerns:**

["NO or VERY MINOR ethics concerns only"]

**Final Justification:**

Thank you for the final comments. I will keep my final score the same and the experiments and novel aspects of the paper are well justified.

**Limitations:**

yes.

**Quality:**

3

**Strengths And Weaknesses:**

Strengths:
1. The adaptive t-norm selection provides a good balance of exploration vs exploitations, giving the model a good chance at choosing the right t-norm for the given scenario.
2. The authors have conducted thorough experimentation and have provided good insights into their finding.
3. The experiments show clear improvement compared to the baseline which is the same model without the constrained loss.
4. Good mathematical support provided by the theoretical guarantees.

Weakness:
1. In terms of novelty, the adaptive t-norm selection is the main novel item, while the constrained loss or the scheduler concepts are not that novel in general in deep learning. MOD-ECL is a novel framework, but the core model part is MOD-YOLO which is not the novelty of this paper. The addition of the adaptive t-norm is the main change to MOD-YOLO.
2. There is no comparison with SOTA on the given datasets, which might help support or deny the need for the motivation of adaptive t-norm selection. The only comparison is with vanilla MOD-YOLO.
3. There are a lot of hyper-parameters, making the tuning part the main difficulty in using this method.
4. The experimental results are not very conclusive on the t-norm selection itself. As pointed by the authors, different t-norms suite different timelines in the training, raising a question, whether a single t norm selection is the right way to go.

---

> ### Author Rebuttal · Authors · 2025-07-31
>
> We thank Reviewer for the thoughtful and thorough critique.
>
> - **In terms of novelty, the adaptive t-norm selection is the main novel item, while the constrained loss or the scheduler concepts are not that novel in general in deep learning. MOD-ECL is a novel framework, but the core model part is MOD-YOLO which is not the novelty of this paper. The addition of the adaptive t-norm is the main change to MOD-YOLO.**
>   **A:** This article focuses on the selection of t-norms to guide the learning process, expanding upon the base of MOD-CL.
>   As the reviewer has stated, adaptive t-norm selection is one key contribution of our work. However, we believe that while the concept of constrained loss and scheduler are not particularly new, the combination of the two is novel.
>   Furthermore, the original paper for MOD-CL does only a shallow exploration of t-norms and their potential use.
>   We have addressed these issues, and have done extensive expansion upon MOD-CL:
>   - We expand the set of t-norms that can be used from 3 to 14, including non-parameterised t-norms.
>   - We expand the evaluation backbones to the latest YOLO model, YOLO11.
>   - We consider two recent and challenging datasets with constraints, ROAD-R and ROAD-Waymo-R.
>
>   MOD-ECL represents not only a conceptual evolution, but also robust and broad empirical research on the integration of constraints into neural networks.
>
> - **There is no comparison with SOTA on the given datasets, only with vanilla MOD-YOLO.**
>   **A:**
>
>   In the context of constrained autonomous driving, MOD-CL is recognised as the state-of-the-art framework for integrating logical requirements into object detection, having won Task 2 of the ROAD benchmark challenge. It extends MOD-YOLO, a multi-label object detection architecture, by incorporating constraint-based loss to enforce semantic compliance with domain rules.
>   While MOD-YOLO provides the underlying detection backbone, it does not support constraint integration by itself.
>   MOD-ECL builds directly on this foundation, enhancing it with adaptive t-norm selection and λ scheduling to improve both constraint satisfaction and detection performance.
>
>   Additionally, our paper compares not only against MOD-ECL baselines that do not use constrained loss, but also against other constrained SOTA methods such as 3D-RetinaNet and I3D (cf. Table 2), as reported in the original ROAD and ROAD-Waymo literature.
>   We stress that the focus of this work is on the integration of logical constraints during training via constrained loss. Accordingly, our experiments are conducted on the constrained variants of the datasets, ROAD-R and ROAD-Waymo-R, which were specifically designed to evaluate such methods.
>
>   Broader comparisons with methods outside the constrained loss paradigm, e.g., transformer-based detectors, are outside the scope of this work.
>   These models typically do not natively support structured constraints, and adapting them would involve altering their training phase, which goes beyond our scope.
>
> - **The method involves many hyper-parameters, making tuning difficult.**
>   **A:** This is precisely what motivates our adaptive approach. By automatically selecting appropriate t-norms during training, the method reduces the need for manual tuning, and helps discover potentially useful configurations.
>   We also perform a study of the parameters in our article, which becomes an indicator towards what t-norms may be of use for training using constrained loss in the autonomous driving setting.
>
> - **Experimental results on t-norm selection are not conclusive; different t-norms suit different phases, questioning whether single t-norm selection is optimal.**
>   **A:** Effectively, our results show that no single t-norm is optimal, but depends on the trade-off between accuracy and constraint satisfaction.
>   For example, Gödel minimises violation count while severely reducing mAP, Łukasiewicz favours mAP, disregarding constraint satisfaction, while Product strikes a balance between the two. See Appendix B for a visualisation.
>   This underpins the usefulness of our adaptive algorithm, as it can automatically guide the search of well performing t-norms with respect to some metric.
>
> - **In section 3.1, it would be good to give some examples for (o, l, a).**
>   **A:** We will include a concrete example to clarify the tuple structure.
>   For example `({Pedestrian}, {Crosswalk}, {Walking, PushObj})` would be one detection, meaning "The pedestrian is on a crosswalk, and is walking while pushing an object".
>
>   It should be noted that each of o, l, a, can be a conjunction of labels, represented via a subset as in the example, as the datasets we use allow multiple of the same type of labels to be present at the same time.
>
> - **Equation 2 would be easier to understand if it is supported by real examples of constraints and t-norms.**
>   **A:** To clarify the semantics of Equation 2, consider the following example constraint:
>
>   > *"If a pedestrian is at a crosswalk, they must be walking."*
>
>   Using standard CNF, this is encoded as:
>
>   ```
>   ¬Pedestrian ∨ ¬Crosswalk ∨ Walking
>
>   ```
>
>   Suppose the model assigns the following confidence values:
>   - Pedestrian: `a = 0.9`
>   - Crosswalk: `b = 0.8`
>   - Walking: `c = 0.7`
>
>   Using the Product t-norm, the constraint violation is:
>
>   ```
>   1 - T_P(a, b, 1-c) = 1 - (a × b × (1-c)) = 1 - (0.9 × 0.8 × 0.3) = 1 - 0.216 = 0.784
>   ```
>
> - **What is the significance of the multi-label part of MOD-ECL? Does it need to be multi-label?**
>   **A:** Yes. Each bounding box in ROAD-R and ROAD-Waymo-R is associated with multiple labels (e.g., object class, location, and action), necessitating a multi-label object detector. This is because one detection can contain multiple labels of the same class type, as shown in the example for (o, l, a) above.
>   We can not use single label classification, even for each class type, hence the need for multi-object detection. (Encoding technically all subsets of labels as the value domain of an auxiliary class will lead to an exponential blowup and is not feasible.)
>
> - **Why not use multiple t-norms for inference?**
>   **A:** The t-norms are used only during training, for the computation of the constrained loss, but not for inference (which is done by the neural network).
>
> - **Merging section 6 into 5.3 might be more intuitive to the reader.**
>   **A:** We agree. We will restructure the sections accordingly in the camera-ready version if accepted to improve clarity and flow.

---

> > ### Comment · Reviewer_DvBY · 2025-08-07
> >
> > 1. On the point of novelty, I agree with the authors that they have made expansions to the MOD-CL model and also used concepts of constrained loss along with scheduling. But my main point is that though these are modifications and additions done in this paper, the main novelty is the adaptive t-norm selection algorithm.
> >
> > 2. Thank you for pointing out that Table 2 includes comparison with 3D-RetinaNet and I3D. I had missed that. Maybe emphasizing this somewhere in the Results section, other than the table description might be useful.
> >
> > 3. The adaptive approach is for the selection of the t-norms so as to avoid choosing the right t-norm based on the dataset. But the algorithm itself includes a lot of hyper parameters like β, λ, δ and γ which affect the performance of the algorithm and the network.
> >
> > Everything else looks good.

---

> > > ### Author Response · Authors · 2025-08-07
> > >
> > > We thank the reviewer for the helpful clarifications and would like to
> > > comment as follows:
> > >
> > > **On the point of novelty**
> > >
> > > We agree that adaptive t-norm selection is the main conceptual novelty, but emphasise that all components are essential to the success of MOD-ECL. The λ-scheduler (cf. Fig. 1) is equally important for proper constraint regularisation over training, and the expansion to a broad space of t-norms enables a more expressive and tunable loss landscape.
> > >
> > > While these additional components may be individually known, they have not previously been combined or systematically evaluated in a unified framework. MOD-ECL is, to our knowledge, the first to integrate these ideas into a coherent, scalable, and fully automated pipeline for constraint-aware training. This integration supports generalisation across model backbones, datasets, and symbolic regimes, and lays the groundwork for further research in structured neurosymbolic learning.
> > > We consider this a significant advance for the state of the art in constrained object detection.
> > >
> > > **On the comparison with 3D-RetinaNet and I3D**
> > >
> > > We will revise Section 5.3 to clarify that Table 2 includes comparisons with 3D-RetinaNet and I3D, the original baselines from ROAD and ROAD-Waymo.
> > >
> > > **On the role of the hyperparameters (β, δ, λ, γ) on the adaptive algorithm**
> > >
> > > The adaptive algorithm introduces only two hyperparameters, β and δ, which govern exploration and score discounting, respectively.
> > > As the adaptive method is fully independent of the λ-scheduler, no γ parameter is needed in this case.
> > >
> > > Crucially, full exploration of static t-norm and λ combinations often requires more runs than the adaptive alternative.
> > > For instance, with 12 t-norms and 3 λ values (e.g., λ = 1, 50, 100), one would need 12 × 3 = 36 runs.
> > > In contrast, with 5 selected (β, δ) pairs (as in Table 3(a)) and the same λ settings, only 5 × 3 = 15 runs are needed.
> > >
> > > Thus, for an initial exploration of t-norm configurations, the adaptive method offers a substantially reduced search space and lower tuning burden.

---

### Official Review · Reviewer_bCQh · 2025-07-01

**Clarity:** 2
**Significance:** 2
**Originality:** 3
**Rating:** 4
**Confidence:** 1

**Summary:**

This paper addresses the challenge of integrating logical constraints into object detection models for autonomous driving to enhance safety. It argues that prior work relies on a static selection of t-norms to formulate a "constrained loss," which can be suboptimal. To solve this, the authors propose MOD-ECL, a neuro-symbolic framework featuring two main contributions: an adaptive algorithm that dynamically selects the most effective t-norm during training, and a scheduler that adjusts the weight of the constrained loss. Through extensive experiments on the ROAD-R and ROAD-Waymo-R datasets, the paper demonstrates that this dynamic approach significantly reduces constraint violations, and in some cases, also improves detection accuracy, enabling fine-grained control over the trade-off between performance and rule compliance.

**Questions:**

See weaknesses.

**Ethical Concerns:**

["NO or VERY MINOR ethics concerns only"]

**Final Justification:**

This article has done solid work in the field of constraint-aware multi-label detection.

**Limitations:**

yes

**Quality:**

3

**Strengths And Weaknesses:**

### Strengths

The paper addresses a significant and practical problem in autonomous driving: ensuring object detection models adhere to real-world logical rules to improve safety. The use of intuitive examples (e.g., a pedestrian on a moving vehicle) effectively highlights the importance of the work. The paper proposes a novel adaptive algorithm for dynamically selecting the best-performing t-norm during training, which is a clever application of multi-armed bandit principles. The paper introduces a λ-scheduler to control the influence of the constrained loss over time, based on the sensible intuition that the model should first learn detection and then focus on constraint satisfaction. The proposed MOD-ECL framework is presented as a clear and modular system for integrating logical constraints. The diagrams and descriptions make the architecture easy to understand.

### Weaknesses

The paper's proposed t-norm method and logic-constrained loss are only compared within the YOLO framework, which demonstrates that different t-norms and λ values indeed influence performance. However, it might be more compelling to compare them against current state-of-the-art (SOTA) methods.

Additionally, multi-modal large language models have shown remarkable performance in scene understanding and reasoning tasks, with a growing number of works extending these models for object detection tasks. It may be beneficial to include comparisons against these approaches to more effectively highlight the advantages of the proposed methods.

---

> ### Author Rebuttal · Authors · 2025-07-31
>
> We thank Reviewer bCQh for the helpful comments and suggestions.
>
> - **- No specific question was asked, though the reviewer suggests comparison against (1) current SOTA methods and (2) multi-modal language models.**
>   **A:** As for (1), MOD-CL is the SOTA for constraint-aware multi-label detection on the ROAD and ROAD-Waymo-R datasets. We also compare against strong competitors such as 3D-RetinaNet and I3D, cited in the original ROAD benchmark.
>   (2) Comparing our framework with larger-scale foundation models or vision transformers is interesting, but was beyond the scope of this work due to their lack of direct support for constrained loss-based learning, as well as multi-label object detection in general.
>
>   In general, object detection models built for autonomous driving train directly on the data without any knowledge of real life constraints. While prior research has shown that their performance for accomplishing the task of object detection is very high, we also observe that these models can and will output predictions that contradict known common-sense rules.
>   To this end, ROAD-R and ROAD-Waymo-R provide requirements that the outputs must satisfy, in the form of logical constraints.
>   Our framework offers seamless integration of these constraints, guiding the training to comply with them, in order to achieve a safer model compared to alternative methods.

---

> > ### Comment · Reviewer_bCQh · 2025-08-04
> >
> > Thanks to the authors' response, I will improve the final score. However, I mentioned multimodal large models for object detection to understand the differences between the best performance achievable by the methods proposed in the paper and the perception of multimodal large models. For example, what is the relationship between the two in terms of metrics and efficiency? This may provide valuable insights for the future development of related fields.

---

> > > ### Author Response · Authors · 2025-08-05
> > >
> > > We thank the reviewer for raising the comparison with multimodal large language models (MLLMs) and for improving the score.
> > >
> > > - In terms of metrics, we do not have f-mAP@0.5 nor MPBV scores for MLLMs in our setting to do a direct comparison.
> > > A possibility is that MLLMs may perform better than the original YOLO model in terms of constraint satisfaction due to the general knowledge of the world they possess.
> > > However, as they do not directly integrate the knowledge of the context-dependent constraints in the datasets we use, we think that MLLMs will still not be able to surpass MOD-ECL in terms of MPBV.
> > > - In terms of efficiency, MOD-ECL performs as fast as YOLO, whereas MLLMs typically require far more computational power. This makes the latter infeasible to use as detectors in real-life settings such as autonomous driving.
> > >
> > > Nonetheless, adapting MLLMs to incorporate constraints directly into their training phase represents an interesting direction for future research.
> > > For a comprehensive overview of recent MLLM-based approaches to perception in autonomous driving, please see "A Survey on Multimodal Large Language Models for Autonomous Driving" by Cui et al. (2024).

---

### Official Review · Reviewer_g3uF · 2025-07-02

**Clarity:** 3
**Significance:** 3
**Originality:** 3
**Rating:** 4
**Confidence:** 3

**Summary:**

Since T-norm selection is a critical factor in detection systems with logical constraints to ensure safety, this paper explores the relationship between T-norm selection and model performance. To address the risk of suboptimal or unstable T-norm choices, the authors conduct a systematic search and analysis for T-norm selection. Building on these findings, they propose the MOD-ECL neuromyotonic framework. This framework dynamically and adaptively selects high-performing T-norms and regulate constrained loss function during training.

**Questions:**

1.What is the computational cost associated with the t-norm selection process? It would be beneficial to report experimental results comparing training time or epoch-level cost with and without t-norm selection.

2.What are the optimal results achieved for subset-based t-norm selection discussed in section 5.1? A more detailed analysis of performance across different t-norm subsets would strengthen the validity of the proposed approach.

**Ethical Concerns:**

["NO or VERY MINOR ethics concerns only"]

**Final Justification:**

1.This paper is well-written and presents a novel insight on guiding t-norm selection for object detection with logical constraints.

2.Extensive experiments on the ROAD-R and ROAD-Waymo-R datasets support the effectiveness and robustness of the proposed t-norm selection.

3.The release of well-documented open-source code and logs enhances the reproducibility and transparency of the work.

Since the authors effectively address my concerns, i will maintain my positive evaluation.

**Limitations:**

Yes

**Quality:**

3

**Strengths And Weaknesses:**

**Strengths**:

1.This paper is well-written and presents a novel insight on guiding t-norm selection for object detection with logical constraints.

2.Extensive experiments on the ROAD-R and ROAD-Waymo-R datasets support the effectiveness and robustness of the proposed t-norm selection.

3.The release of well-documented open-source code and logs enhances the reproducibility and transparency of the work.

**Weaknesses**:

1.There are some little typos, like the mod-ec in Line 156, and the organization of appendix materials.

2.The presentation of paper could be better. For instance, it is hard to follow the difference between the proposed framework MOD-ECL and based framework MOD-CL. Meanwhile, although 5.1 propose a methodology, it is not explained how a subset of t-norm selection chooser and scheduler choose.

3.The contribution of MOD-ECL framework compared with MOD-CL should be clarified in both experiment results sections and method sections.

---

> ### Author Rebuttal · Authors · 2025-07-31
>
> We thank the Reviewer for the comments and insightful feedback.
>
> - **The presentation of paper could be better. For instance, it is hard to follow the difference between the proposed framework MOD-ECL and based framework MOD-CL.
> The contribution of MOD-ECL framework compared with MOD-CL should be clarified in both experiment results sections and method sections.**
>  **A:** This article focuses on the selection of t-norms to guide the learning process.
>   MOD-CL serves as a basis to explore this, as the implementation already could handle multi object detection and constrained loss.
>   However, the original MOD-CL paper neither experiments with nor provides a detailed discussion on the impact of different t-norm choices or the role of regularisation techniques.
>
>   The novel conceptual contributions of MOD-ECL are as follows:
>   - An adaptive algorithm that automatically guides the search of well performing t-norms.
>   - A scheduler for constrained loss regularization.
>   - A metric for measuring constraint satisfaction.
>
>   Experimentally, our contributions are as follows:
>   - We expand the set of t-norms that can be used from 3 to 14, including non-parameterised t-norms.
>   - We expand the evaluation backbones to the latest YOLO model, YOLO11.
>   - We consider two recent and challenging datasets with constraints, ROAD-R and ROAD-Waymo-R.
>
> - **What is the computational cost associated with the t-norm selection process? It would be beneficial to report experimental results comparing training time or epoch-level cost with and without t-norm selection**
>   **A:** We have repeated a representative subset of key configurations (baseline, static product t-norm and adaptive) three times and now report the mean, standard deviation, and 95% confidence intervals, alongside wall-clock time, FLOPs and peak GPU memory usage.
>   We stress that at inference time, there is no computational overhead.
>   Please see the reply to Reviewer 0bpq.
>
> - **What are the optimal results achieved for subset-based t-norm selection discussed in section 5.1? A more detailed analysis of performance across different t-norm subsets would strengthen the validity of the proposed approach**
>   **A:**  Although we acknowledge the importance of analysing the performance of different t-norm subsets, in this work we deliberately evaluate using the full set of 14 t-norms to avoid introducing selection bias and to ensure a fair comparison across configurations. Exhaustively testing all possible subsets would be computationally infeasible and statistically ambiguous due to the combinatorial explosion (16383 subsets). Moreover, selecting a few specific subsets bears the risk of introducing arbitrary choices that may not generalise or align with domain-agnostic evaluation. Instead, our aim in this work is to assess the effectiveness of adaptive selection across the complete space of standard t-norms. That said, the framework is designed to support arbitrary t-norm subsets and domain-specific prior selection, which we leave as a direction for future targeted studies.

---

> > ### Comment · Reviewer_g3uF · 2025-08-04
> >
> > Thank you for your response and for conducting the additional experiments. They effectively address my concerns. I appreciate your efforts and will maintain my positive score.

---

### Official Review · Reviewer_qbpq · 2025-07-05

**Clarity:** 3
**Significance:** 2
**Originality:** 3
**Rating:** 4
**Confidence:** 2

**Summary:**

The paper proposes MOD-ECL, a neurosymbolic framework that enhances multi-label object-detection networks for autonomous driving by incorporating logical-constraint supervision through three key innovations: a broadened library of t-norms (standard, non-parameterized, and parameterized) for translating first-order rules into differentiable losses, an adaptive t-norm selection algorithm that treats the choice as a multi-armed bandit problem with discounted reward updates, and a λ-scheduler that gradually increases the weight of constrained loss to prioritize object detection learning before rule compliance. Experiments on ROAD-R and ROAD-Waymo-R benchmarks using YOLOv8n/11n backbones demonstrate that careful t-norm and λ selection can reduce mean per-box violations by up to ~30% and occasionally improve frame-mAP by a few points compared to unconstrained baselines, with the adaptive algorithm favoring parameterized Yager t-norms while the scheduler provides controllable trade-offs between accuracy and compliance.

**Questions:**

- Could the authors run at least 3–5 repetitions using different seeds and report mean ± standard deviation or 95% confidence intervals for mAP and MPBV?

- The expanded t-norm set and adaptive algorithm may introduce additional computational costs. Please quantify the GPU memory usage, FLOPs, and wall-clock time overhead compared to baseline and previously published constrained-loss methods.

- Currently, the exponential λ-scheduler is only compared against a static λ. Could alternative schedules (e.g., linear, cosine) or an automatically tuned final λ value be explored, and the sensitivity of the trade-off curve discussed?

- The paper hypothesizes potential applicability to healthcare and other domains (§7). Could a preliminary experiment with multilabel medical images incorporating simple anatomical constraints be included to demonstrate transferability?

**Ethical Concerns:**

["NO or VERY MINOR ethics concerns only"]

**Final Justification:**

Thank you for all the responses and comments. I will keep my final score the same

**Limitations:**

Yes

**Quality:**

3

**Strengths And Weaknesses:**

Strengths

- The formulation of the constrained loss, adaptive t-norm selection, and λ-scheduler is well-defined, and an upper-bound regret theorem is provided.

- The experiments comprehensively cover 14 different t-norms, two datasets, and two distinct model sizes.

- The paper's structure follows a logical progression (background → method → experiments), enhancing readability and comprehension.

- Figures effectively illustrate the trade-offs between constraint violation and accuracy (Fig. 1, Fig. 3), while the inclusion of pseudocode significantly aids reproducibility.

Weaknesses

-  The statistical rigor of the experiments is limited. All results presented are from single experimental runs, without reporting confidence intervals or statistical significance tests (as noted in Checklist Q7). Additionally, the memory and computational time overhead associated with the expanded set of t-norms has not been quantified.

-  The reported improvements in mAP scores are modest.

---

> ### Author Rebuttal · Authors · 2025-07-31
>
> We thank Reviewer for the feedback and valuable suggestions.
>
> - **The reported improvements in mAP scores are modest.**
>   **A:** When comparing our model's mAP scores to existing baselines (the first column of each set in Table 2), we see that the improvements are significant. In some cases, the performance nearly doubles the best existing work, indicating the strength of our framework in general. Moreover, when comparing to our given baselines, our t-norm implemented models also have increase in performance (+3.45 for the biggest difference).
>   This margin is considerable, given that architectural improvements (YOLOv11 vs. v8) typically yield smaller gains (e.g., +1.3 mAP).
>
>   However, we emphasise that our main goal is not to increase performance in terms of mAP sorce or similar parameters, but to improve the compliance of the model with the semantic constraints, without significant cost in such performance.
>   Our results show that this goal is achieved, and that in fact performance might be even improved.
>
> - **Could the authors run at least 3–5 repetitions using different seeds and report mean ± standard deviation or 95% confidence intervals for mAP and MPBV?**
>   **A:** Due to the significant computational cost of full-scale evaluation, repeating all experiments is infeasible in short time. However, we have repeated a representative subset of key configurations (baseline, static product t-norm and adaptive) three times and now report the mean, standard deviation, and 95% confidence intervals. Unfortunately, due to time and resource constraints, we were only able to execute the adaptive configuration once with YOLOv8n, and twice with YOLO11n.
>   The results support the consistency of our findings: variance is generally low, with standard deviations typically below 1.0 for mAP and around 1.0–1.3 for MPBV. For instance, on ROAD-R with yolov8n, the adaptive setup achieved 59.02 ± 0.8 mAP and 36.29 ± 0.9 MPBV, compared to the original reported values of 58.93 mAP and 36.34 MPBV in Table 3a.
>   This suggests that both detection performance and constraint satisfaction remain stable across different seeds. Moreover, the observed improvement of up to +3.45 mAP between baseline and constrained configurations (cf. 56.34->60.77 for yolov8n on ROAD-R) is larger than the gap between model sizes (e.g. YOLO11 baseline vs. next-best, +1.3), reinforcing that the gains are meaningful rather than noise.
>
>
> - **The expanded t-norm set and adaptive algorithm may introduce additional computational costs. Please quantify the GPU memory usage, FLOPs, and wall-clock time overhead compared to baseline and previously published constrained-loss methods.**
>   **A:** We now provide a comparative breakdown of computational overheads introduced by adaptive and static t-norm selection. These include GPU memory usage, FLOPs, total training time, and per-epoch wall-clock time.
>   Our method incurs no additional overhead at inference time, as t-norms are used exclusively during training.
>
>   **Summary of Runs:**
>   *Mean ± standard deviation and 95% confidence intervals for mAP and MPBV over three seeds (0,1,2) for each configuration, with the exception of adaptive YOLOv8n and YOLO11n on ROAD-Waymo-R, where the former is only run with seed 0 and the latter with seeds 0,1. Also reported are total training time, FLOPs, and peak GPU memory usage.*
>
>     | **Dataset**     | **Base Model** | **Config**  | **mAP** | **MPBV** | **Train Time (s)** | **FLOPs (G)** | **Peak Mem (GB)** |
>     |-----------------|----------------|-------------|--------:|---------:|-------------------:|---------------:|------------------:|
>     | ROAD-R          | yolov8n        | Baseline               | 56.34 ± 0.6 (CI 1.98) | 41.18 ± 1.3 (CI 2.0) | 1176 ± 31.96 | 4.12 | 71.38 ± 3.28 |
>     |                 |                | Product (λ=50)         | 60.77 ± 1.0 (CI 2.61) | 36.56 ± 1.2 (CI 1.9) | 2027 ± 30.72 | 4.12 | 73.25 ± 0    |
>     |                 |                | Adaptive (β=0.5,γ=0.5) | 59.02 ± 0.8 (CI 2.54) | 36.29 ± 0.9 (CI 1.5) | 2800 ± 87.93 | 4.12 | 73.25 ± 0    |
>     |                 | yolov11n       | Baseline               | 61.69 ± 0.8 (CI 2.03) | 40.18 ± 3.4 (CI 6.6) | 1481 ± 32.58 | 3.24 | 77.24 ± 0 |
>     |                 |                | Product (λ=50)         | 58.06 ± 1.0 (CI 2.50) | 36.38 ± 2.2 (CI 5.1) | 2319 ± 24.27 | 3.24 | 77.24 ± 0 |
>     |                 |                | Adaptive (β=0.5,γ=0.5) | 57.94 ± 0.9 (CI 2.42) | 34.67 ± 1.4 (CI 3.6) | 3374 ± 26.89 | 3.24 | 77.24 ± 0 |
>     | ROAD-Waymo-R    | yolov8n        | Baseline               | 34.96 ± 0.27 (CI 0.78) | 47.19 ± 0.88 (CI 2.48) | 15874 ± 70.35   | 4.12 | 23.04 ± 0 |
>     |                 |                | Product (λ=100)        | 35.12 ± 0.05 (CI 0.14) | 43.63 ± 0.81 (CI 2.30) | 21258 ± 233.18  | 4.12 | 23.04 ± 0 |
>     |                 |                | Adaptive (β=0.5,γ=0.5) | 33.51 ± 0 (CI 0.00)    | 45.59 ± 0 (CI 0.00)    | 24903 ± 0.00    | 4.12 | 23.04 ± 0 |
>     |                 | yolov11n       | Baseline               | 34.65 ± 0.28 (CI 0.79) | 46.32 ± 0.40 (CI 1.12) | 16254 ± 33.47   | 3.25 | 23.44 ± 0 |
>     |                 |                | Product (λ=100)        | 33.64 ± 0.08 (CI 0.24) | 42.99 ± 1.01 (CI 2.84) | 20157 ± 74.87   | 3.25 | 23.44 ± 0 |
>     |                 |                | Adaptive (β=0.5,γ=0.5) | 32.91 ± 0.42 (CI 1.17) | 43.02 ± 0.53 (CI 1.50) | 25983 ± 1356.45 | 3.25 | 23.44 ± 0 |
>
>
> - **Currently, the exponential λ-scheduler is only compared against a static λ. Could alternative schedules (e.g., linear, cosine) or an automatically tuned final λ value be explored, and the sensitivity of the trade-off curve discussed?**
>     **A:** While we focus on exponential scheduling in this work, the method is compatible with other schedulers.
>     To illustrate this concretely, consider a linear decay alternative to the exponential schedule currently used.
>     The constraint weight decays linearly from `λ_0` to `0` over the remaining training epochs:
>
>
>     ```
>     λ_t = λ_0 × (t / T)
>     ```
>
>     Where:
>     - `λ_0` is the initial constraint weight,
>     - `T > 1` is the total number of epochs.
>
>    Our implementation is able to support such schedulers, and we will clarify citing linear, cosine schedules along adaptive λ-tuning as promising directions for future work.
>
> - **The paper hypothesises potential applicability to healthcare and other domains (S7). Could a preliminary experiment with multilabel medical images incorporating simple anatomical constraints be included to demonstrate transferability?**
>   **A:** While we support the motivation, we are unaware of any publicly available multilabel medical image dataset annotated with formal constraints.
>   Constructing such a dataset would require meticulous crafting of both label structure and constraint definitions, which is a non-trivial undertaking and would demand significant time and domain expertise.
>   Nonetheless, consider an illustrative and simplified example from a surgical scene understanding task:
>   > *Constraint:* “Incision actions require holding a scalpel”
>   Translated as:
>   ```
>   ¬(Incision ∧ ¬Scalpel) ≡ ¬Incision ∨ Scalpel
>   ```
>   This formulation is structurally analogous to our constraints in the autonomous driving domain.

---

> > ### Comment · Reviewer_qbpq · 2025-08-03
> > **Thanks for your response**
> >
> > Thank you for your response and the additional experiments. I will keep my positive score.

---

### Note · Authors · 2025-08-13

While not a pressing comment, we believe it may be useful to provide a brief summary of the outcome of our interactions with the reviewers, for use at the discretion of the chairs.

**Contribution.** We clarified that MOD-ECL’s impact lies in integrating logical constraints into object detection to significantly reduce violation rates (up to ~30 %) without harming—and sometimes improving—accuracy. This is achieved through the unified design of an expanded t-norm library, adaptive selection, and λ-scheduling—an integration not previously implemented or systematically evaluated.

**Evaluation & Cost.** Rebuttal experiments confirmed low variance across seeds and revealed no inference overhead.

**Comparisons.** We clarified inclusion of strong baselines (3D-RetinaNet, I3D) and discussed MLLMs as future work given their current limitations for constraint-aware training.

**Planned Improvements.** If accepted, we will (i) emphasise baseline comparisons in the results section while clearly distinguishing MOD-ECL from MOD-CL, (ii) correct typos and add examples where necessary and (iii) reorganize the sections of the main text and appendix.

The discussions with the reviewers have been fruitful, and the process has helped to better demonstrate MOD-ECL’s ability as a constraint-aware training framework for autonomous driving. We believe our contributions are highly valuable in advancing safer and more reliable vision models.

---

### Decision · Program_Chairs · 2025-09-17

**Decision:**

Accept (poster)

**Comment:**

The paper proposes MOD-ECL, a neurosymbolic framework for constraint-aware multi-label object detection in autonomous driving. MOD-ECL extends prior work (MOD-CL) by: (1) Expanding the set of t-norms from 3 to 14, including parameterized and non-parameterized variants. (2) Introducing an adaptive t-norm selection algorithm, framed as a multi-armed bandit, to dynamically guide training. (3) Adding a λ-scheduler to regulate the influence of constraint loss during different training stages.

Experiments on the ROAD-R and ROAD-Waymo-R benchmarks with YOLOv8/11 backbones show that MOD-ECL reduces mean per-box violations (MPBV) by up to ~30% without inference overhead, and occasionally improves detection accuracy (frame-mAP). The approach provides a principled way to balance accuracy and rule compliance, with theoretical guarantees (upper-bound regret for adaptive selection).

Despite borderline ratings (4× borderline accept), the paper demonstrates solid technical depth, clear motivation, and valuable contributions to constrained object detection. The adaptive t-norm selection and λ-scheduler represent a meaningful step toward safer, constraint-aware perception models. The rebuttal period resolved major concerns on statistical rigor and computational overhead, with reviewers acknowledging the additional experiments and clarifications. All four reviewers maintained borderline-accept ratings, with final justifications leaning positive. Consensus emerged that the work is technically solid and contributes to safer object detection, warranting acceptance.